



**Retrogressive thaw slumps temper dissolved organic carbon delivery to streams of the Peel Plateau,**
**NWT, Canada**
Cara A. Bulger[1*], Suzanne E. Tank[1], and Steven V. Kokelj[2]
[1]Department of Biological Sciences, University of Alberta, Edmonton, AB, Canada, T6G 2E9
[2]Northwest Territories Geological Survey, Government of the Northwest Territories, Yellowknife, NT,
Canada
[*]Author for correspondence: cara.bulger@gmail.com





**Abstract**
In Siberia and Alaska, permafrost thaw has been associated with significant increases in the delivery of
dissolved organic carbon (DOC) to recipient stream ecosystems. Here, we examine the effect of
retrogressive thaw slumps (RTS) on DOC concentration and transport, using data from eight RTS features
on the Peel Plateau, NT, Canada. Like extensive regions of northwestern Canada, the Peel Plateau is
comprised of thick, ice-rich tills that were deposited at the margins of the continental ice sheet. RTS
features are now widespread in this region, with headwall exposures up to 30 m high, and total
disturbed areas often exceeding 30 ha. We find that intensive slumping on the Peel Plateau is universally
associated with decreasing DOC concentrations downstream of slumps, even though the composition of
slump-derived dissolved organic matter (DOM; assessed using specific UV absorbance and slope ratios)
is similar to permafrost-derived DOM from other regions. Comparisons of upstream and downstream
DOC flux relative to a conservative tracer suggest that the substantial fine-grained sediments released
by slumping may sequester DOC on this landscape. Runoff obtained directly from within slump features,
above entry into recipient streams, indicates that the deepest RTS features, which thaw the greatest
extent of buried, Pleistocene-aged glacial tills, have the lowest runoff DOC concentrations when
compared to upstream, un-disturbed locations. In contrast, shallower features, with exposures that are
more limited to a relict Holocene active layer, have within-slump DOC concentrations more similar to
upstream sites. Finally, fine-scale work at a single RTS feature indicates that temperature and
precipitation serve as primary environmental controls on above-slump and below-slump DOC flux, but
that the relationship between climatic parameters and DOC flux is complex for these dynamic
thermokarst features. These results demonstrate that we should expect striking variation in
thermokarst-associated DOC mobilization across Arctic regions, but that within-region variation in
thermokarst intensity and other landscape factors are also important for determining biogeochemical
response. An understanding of landscape and climate history, permafrost genesis, soil composition, the
nature and intensity of thermokarst, and the interaction of these factors, is critical for predicting

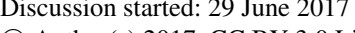



changes in land-to-water carbon mobilization in a warming circumpolar world.
**1. Introduction**

Anthropogenic climate change is significantly affecting the Canadian Arctic cryosphere (IPCC,

2014). Temperature increases in Arctic regions are predicted to be at least 40% greater than the global
mean (IPCC, 2014), while precipitation is also expected to increase significantly (Walsh et al., 2011). The
resulting degradation of permafrost is forecast to have wide-ranging effects, because thawing has the
potential to greatly alter the physical, chemical, and biological functioning of landscapes (Frey and
McClelland, 2009; Khvorostyanov et al., 2008a, 2008b; Kokelj et al., 2017b; Schuur et al., 2008, 2013). In
particular, permafrost acts as a long term storage medium for solutes and sediments, and as a barrier to
the participation of permafrost-sequestered constituents within active biogeochemical cycles (McGuire
et al., 2009). Consequently, permafrost thaws enhances linkages between terrestrial and aquatic
systems, via increasing transport of terrestrial compounds from land to water (Kokelj et al. 2013; Tanski
et al., 2016; Vonk et al., 2015b). Given that global permafrost stores of carbon are estimated to be
almost double that of the atmospheric carbon pool (Hugelius et al., 2014), there is great potential for
large increases in carbon mobilization as a result of permafrost thaw (Schuur et al., 2015). Within this
context, the transport of dissolved organic carbon (DOC) from land to water is of particular interest,
because DOC acts as the primary substrate for the microbially-mediated mineralization of organic
carbon to carbon dioxide (Battin et al., 2008; Spencer et al., 2015). Dissolved organic carbon also forms
the majority of total organic carbon flux in most Arctic rivers (Spencer et al., 2015), and is thus the
primary vehicle for the delivery of terrestrial carbon to the Arctic Ocean (Dittmar and Kattner, 2003;
Holmes et al., 2012). As a result, the implications of thaw-mediated DOC mobilization may range from
effects on the permafrost-carbon feedback, to the ecological and biogeochemical functioning of
streams, rivers, and the nearshore ocean (e.g. Tank et al., 2012b; Vonk et al., 2015b).



Permafrost thaw can manifest in many different forms, ranging from an increase in active layer
thickness and terrain subsidence, to thermokarst features that significantly reconfigure the physical
structure of the landscape. Of these, thermokarst has the potential to rapidly expose significant
quantities of previously-frozen soils to biological and chemical processing (Abbott et al., 2014, 2015;
Kokelj and Jorgenson, 2013; Malone et al., 2013). One of the most conspicuous manifestations of
thermokarst is the retrogressive thaw slump (RTS), which develops in ice-rich glacial deposits across
northwestern Canada, Alaska, and western Siberia (Kokelj et al., 2017b), and in Yedoma regions of
Alaska and Siberia (Murton et al., 2017). Thaw slumps are widespread throughout glaciated terrain in
the western Canadian Arctic (Kokelj et al., 2017b), including on the Peel Plateau (Lacelle et al., 2015).
These dynamic landforms develop via the ablation of an ice-rich headwall and – through the coupling of
geomorphic and thermal processes – are particularly efficient at thawing thick zones of ice-rich
permafrost and translocating large volumes of sediment from slopes to downstream environments (see
Fig. 1). RTS features remain active for decades. They typically stabilize following sediment accumulation
at the base of the headwall that insulates the ground ice and arrests thaw (Kokelj et al., 2015), but can
reactivate causing thaw within the scar zone, and upslope expansion of the disturbance (Kokelj et al.,
2013; Lantuit and Pollard, 2008). During periods of activity, thawed materials accumulate as a saturated
slurry in the slump scar zone (see Fig. 1b) and are translocated downslope by mass flow processes,
which are accelerated by meltwater- and rainfall-induced saturation (Kokelj et al. 2015). During active
and stabilized periods, surface runoff can also remove solutes and suspended sediment from the
thawed substrate to downstream environments. Although variation in temperature, precipitation and
solar radiation have been correlated with development rates and growth of retrogressive thaw slumps
(Kokelj et al., 2009, 2013, 2015; Lacelle et al., 2010; Lewkowicz, 1986, 1987), we know little about how
these and other environmental drivers might control permafrost-DOC dynamics at the individual-slump
to small watershed scale.



On the Peel Plateau, individual thaw slumps commonly impact tens of hectares of terrain,
displace hundreds of thousands of cubic meters of sediments downslope, and significantly alter surface
water sediment and solute loads (Kokelj et al., 2013; Malone et al., 2013), and thus downstream
ecosystems (Chin et al., 2016; Malone et al., 2013). The magnitude of these disturbances and their
cumulative impacts is great enough to alter solute loads in the Peel River (70,000 km$^2$ watershed area;
Kokelj et al., 2013), even though only a small portion of the river's total catchment area (<1%) is
influenced by thermokarst (Kokelj et al., 2017b; Segal et al., 2016). This contrasts with many other
permafrost-affected regions, where increases in solute loads following permafrost disturbance can be
transient (e.g., limited to spring freshet) and have little overall effect on annual solute fluxes (for
example, in High Arctic regions affected by active layer detachments; Lafrenière & Lamoureux, 2013). In
addition, permafrost thaw on the Peel Plateau is notable in that it exposes vast quantities of mineral-
rich glacial till, which is overlain by a relatively shallow layer of slightly more organic-rich soils (Duk-
Rodkin and Hughes, 1992; Kokelj et al. 2017a). Although this landscape type is found across glaciated
permafrost terrains of the circumpolar North (e.g., Kokelj et al. 2017b), it contrasts with regions of
Alaska and eastern Siberia that are either Yedoma-rich or were patchily glaciated during the late
Pleistocene, and which have been common focus points for study of permafrost-DOC interactions to
date (Abbott et al., 2014, 2015; Drake et al., 2015; Mann et al., 2012; Vonk et al., 2013b).
In several Arctic regions, permafrost thaw, including thermokarst, has been documented to
enhance DOC concentrations in recipient aquatic ecosystems (Frey and McClelland, 2009; Tank et al.,
2012a; Vonk et al., 2013a; Vonk and Gustafsson, 2013). For example, streams draining thaw slumps have
higher DOC concentrations than un-affected systems across various terrain types in Alaska (2-3 fold
increase; Abbot et al., 2014), while the DOC concentration in runoff from thawing Yedoma in eastern
Siberia is considerably greater than concentrations in recipient river systems (~30-fold elevation;
Spencer et al. 2015). However, multiple factors, including variable carbon content in permafrost soils
(Hugelis et al. 2014) may affect DOC release with permafrost thaw. In regions where thermokarst



transports fine-grained sediments to aquatic systems, sorption processes may also be important,
because dissolved organic matter (DOM) can readily sorb to mineral soils. This rapid process is largely
regulated by the chemical composition and clay content of mineral sediments (Kothawala et al. 2009),
and can cause DOM to be rapidly removed from solution in stream systems (Kaiser and Guggenberger,
2000; McDowell, 1985).The DOM-mineral complex can be an important mechanism for enabling the
downstream transport and continued sequestration of organic carbon (Hedges et al., 1997). Sorption
processes may be particularly important for DOC transport in the glaciated western Canadian Arctic,
where landscape predisposition to thaw slumping results in an abundance of thermokarst related slope
disturbances which effectively mobilize fine-grained glacial sediment stores to downstream systems
(Kokelj et al., 2017a, 2017b; Rampton, 1988).

In this study, we quantify how RTS features on the Peel Plateau affect the concentration and

composition of DOC within a series of recipient stream systems, to explore how DOC mobilization from
land to water is affected by thermokarst in this region. We further investigate how short-term variation
in precipitation, temperature, and solar radiation affect DOC flux above and below a single RTS feature,
to explore the drivers of temporal variation in DOC flux. We specifically target these thermokarst-
sensitive glacial deposits, which are characteristic of large portions of the circumpolar Arctic, to explicitly
consider how variations in permafrost soil composition, permafrost genesis, and Quaternary history,
influence variability in permafrost-DOC interactions across vast Arctic regions. The study results broaden
our understanding of land-water carbon mobilization in permafrost terrain, and indicate that slumping
on the Peel Plateau may act to temper the flux of DOC within this landscape, via mineral-carbon
interactions. These findings also underline the importance of landscape characteristics and geological
inheritance for determining the biogeochemical effects of thermokarst, particularly as hillslope
thermokarst intensifies across many Arctic regions (Kokelj et al., 2017b).




## 2 Study Site

### 2.1 General study site description

Our study was conducted on the Peel Plateau, situated in the eastern foothills of the Richardson

Mountains, NWT, Canada, in the zone of continuous permafrost (Fig. 1a) (Kokelj et al., 2016). The

fluvially-incised Plateau ranges in elevation from 100 to 650 masl (Catto, 1996). The region was covered

by the Laurentian Ice Sheet (LIS) for a brief period 18,500 cal yr BP (Lacelle et al., 2013). The bedrock of

the region is Lower Cretaceous marine shale from the Arctic River formation (Norris, 1984) and siltstone

overlain by Late Pleistocene glacial, glacio-fluvial and glacio-lacustrine sediments (Duk-Rodkin and

Hughes, 1992), covered by a shallow organic layer. These Pleistocene deposits host ice-rich permafrost.

Carbon dating and $^{18}O$ measurements in the region have placed the age of relict ground ice in the late

Pleistocene epoch (18,100 ± 60 $^{14}C$yr BP; Lacelle et al., 2013). Upper layers of permafrost thawed during

the early Holocene and host younger, Holocene-aged organic materials (7890 ± 250 $^{14}C$yr BP; Lacelle et

al., 2013). These are clearly delineated from deeper Pleistocene-aged permafrost by a thaw

unconformity, which developed when warmer climate during the early Holocene prompted the thawing

of near-surface permafrost and a regional increase in active layer thicknesses, enabling the leaching of

soluble ions and integration of organic matter into these previously thawed soils (see Fig. 1c-d).

Subsequent aggradation of permafrost due to gradual cooling has archived this notable stratigraphic

variation in geochemistry, organic matter content, and cryostructure (Kokelj et al., 2002; Lacelle et al.,

2014; Murton and French, 1994).

Ice-marginal glacigenic landscapes such as the Peel Plateau host thick layers of ice-rich

sediments, and thus have a predisposed sensitivity to climate-driven thaw slump activity (Kokelj et al.,

2017). On the Peel Plateau, slumping is largely constrained by the maximum extent of the LIS, because

the thick layers of ice-rich permafrost necessary for RTS activity is not present beyond its glacial limits

(Lacelle et al., 2015). Fluvial incision provides the topographic gradients necessary for thaw slump





development and RTS features are common; ranging in size from small, newly developing features,
which are relatively numerous, to those greater than 20 ha, which are rare (<5% prevalence; Lacelle et
al., 2015). The recent intensification of slumping on the Peel Plateau is driven in part by increasing air
temperatures and summer rainfall intensity (Kokelj et al., 2015). This intensification is also increasing the
thaw of the deepest layer of ice-rich, organic-poor, Pleistocene-aged glacigenic tills that underlie this
region. The pattern of abundant thaw slump development across ice-marginal glaciated permafrost
landscapes extends from the Peel Plateau across the western Canadian Arctic, and persists at
continental scales (Kokelj et al., 2017b).

*2.2 Regional climate*

The regional climate is typical of the subarctic with long, cold winters and short, cool summers.

Mean annual air temperature (1981-2010) at the Fort McPherson weather station (Fig. 1a) is -7.3 $^{o}$C
with average summer (June-August) temperatures of 13.3 $^{o}$C (Environment Canada, 2015). A warming
trend of 0.77 $^{o}$C per decade since 1970 has been recorded; however these increases are most apparent
in the winter months (Burn and Kokelj, 2009). Our sample period spanned the thaw months of July and
August; average 1981-2010 temperatures for those months, recorded at Fort McPherson, are 15.2 and
11.8 $^{o}$C, respectively, slightly higher than averages observed at a more elevated, centrally-located
meteorological station (Fig. 1a) during our study (13.2 $^{o}$C in July and 9.5 $^{o}$C in August). Annual cumulative
rainfall (1981-2010) at the Fort McPherson weather station averages 145.9 mm, with July and August
having the highest rainfall levels at 46.4 and 39.1 mm (Environment Canada, 2015). In 2014, rainfall for
July and August was 128.7 and 170.7 mm. This makes 2014 a cooler year than average, and continues
the trend of increasingly wetter summers with numerous extreme rainfall events (Kokelj et al., 2015).






**3 Methods**

*3.1 Slump site selection*

Eight RTS features were selected from across the study region (Fig. 1; Fig. S1; Table 1). Selected

slumps possessed a debris tongue that extended to the valley bottom and directly impacted a stream

system. Sampling at each slump occurred at three discrete locations: upstream, within-slump, and

downstream of slump influence (Fig. 1b). Upstream sites were trunk streams that connected with the

slump flow path further downstream, and were un-affected by any major geomorphic disturbance and

thus representative of an undisturbed, pristine environment. Within-slump sampling locations were

locations of channelized slump runoff within the scar zone or upper debris tongue. Downstream

sampling locations were located below the confluence of the sampled upstream flow and all within-

slump runoff paths, and were chosen to be representative of slump impact on aquatic ecosystems

across the Peel Plateau landscape. In one instance (Slump HD, August 17), a fluidized flow event

between sampling events saturated the scar zone and obliterated within-slump channelized surface

flow. As a result, the within-slump sample taken at this site was not representative of typical

channelized slump runoff that characterized all other slump sampling conditions, and has been

discarded from all analyses.

A general classification of the slumps is difficult as these features are influenced by a diverse

range of geomorphic processes that vary in intensity over time (Table 1; Fig. S1). Three of the slumps

(FM4, FM2, FM3) are classified as 'mega slumps', characterised by areas greater than 5 ha, a headwall

greater than 4 m in height, and a debris tongue that connects the slope to the valley below (Kokelj et al.,

2013, 2015). Of these, FM4 possesses a substantial headwall approximately 20 m in height, but is

currently largely stabilized, indicated by the small outflow, long, dry, and significantly revegetated debris

tongue (Fig. S1). FM2 is among the largest active slumps in the region, with a headwall 25-30 m high and

visible as a much smaller feature in air photos since 1944 (Lacelle et al. 2015). FM2 geochemistry and



geomorphology were described by Malone et al. (2013). Slump FM3, which was chosen for our
'environmental controls' work (further described below) covers an area of approximately 10 ha, and has
a headwall of approximately 10 m in height and a debris tongue that extends nearly 600 m down valley
(Table 1). Headwall retreat rate at FM3 over a 20 year period has been calculated at 12.5 m yr$^{-1}$ (Lacelle
et al., 2015). SD is the smallest and youngest slump that we studied, and was initiated when diversion of
a small creek caused lateral bank erosion. The SD headwall is 2-4 m high with a scar zone that extends
approximately 20m, and no defined debris tongue. The remaining slump sites (HA, HB, HC, HD) were all
well-developed active RTS features with headwalls similar to, or smaller than, FM3, but with debris
tongues that are much smaller in volume (Table 1). With the exception of SD, slump headwalls exposed
permafrost well below a thaw unconformity, indicating that Pleistocene-aged, unweathered glacigenic
materials are being thawed by the slump (Lacelle et al., 2013).

*3.2 Field sampling and data collection*
3.2.1 The effect of slumping on DOC and stream water chemistry

The majority of our sampling was conducted during the summer of 2014. At each slump,

samples were collected at upstream, downstream, and within-slump locations. Of the eight slumps that
were sampled, three were accessed from the Dempster Highway three times over the sampling season,
one (FM3; see also 3.2.2) was accessed twice from the highway, and four were accessed twice via
helicopter (Table 1). At each sampling location, conductivity, pH, and temperature were recorded using
a YSI Pro Plus multi-parameter meter. Water samples were collected from directly below the stream
surface into 1 L acid washed HDPE bottles and allowed to sit in chilled, dark conditions for 24 hours to
enable the substantial sediments in these samples to settle out of suspension. Sample water was then
filtered with pre-combusted (475$^{o}$C, 4 hours) Whatman GF/F filters (0.7 µm pore size). Filtered sample
water was transferred into 40 mL acid washed, pre-combusted glass bottles for DOC analysis, or 60 mL
acid washed HDPE bottles for the analysis of absorbance and major ions. DOC samples were acidified



with hydrochloric acid (1μL mL$^{-1}$), following Vonk et al. (2015b). All samples were refrigerated until
analysis. The GF/F filters were retained for total suspended sediment (TSS) analysis. Samples for stable
water isotope analyses were collected directly from streams into acid washed 40 mL HDPE bottles with
no headspace. Bottles were sealed and refrigerated until analysis. During 2016, samples were
additionally collected from a subset of slump locations (FM2, FM3, FM4 and SD) for the $^{14}$C signature of
DOC at upstream and within-slump sites. Field samples were collected in pre-washed 1-2 L
polycarbonate bottles, allowed to settle for 24 hours, and filtered using pre-combusted Whatman GF/F
filters into pre-combusted glass media bottles with phenolic screw caps with butyl septa. Sample bottles
were wrapped in aluminum foil and refrigerated until analysis.

3.2.2 Environmental controls on DOC flux
To explore how environmental variables control the flux of DOC from RTS-affected streams, we
visited slump FM3 an additional 17 times beyond the sampling described above. During each visit, we
measured discharge at the upstream and downstream locations to calculate DOC flux, and collected
upstream and downstream DOC concentration samples. Downstream discharge was measured using an
OTT C2 current meter at three locations across the small stream and at 40% depth. Due to the shallow,
low flow conditions at the upstream site, upstream discharge was measured using the cross sectional
method (Ward and Robinson, 2000). In both cases, discharge was calculated as the product of velocity
and stream cross-sectional area. Local daily climate data were obtained from an automated
meteorological station previously established in 2010 by the Government of the Northwest Territories
(Kokelj et al. 2015). The station is located within 2 km of slump FM3 (Fig. 1a) and is instrumented for the
measurement of air temperature, rainfall, and net radiation.






*3.3 Laboratory analyses*
3.3.1 Major ions, dissolved organic carbon, $\delta^{18}O$ and $DO^{14}C$

Cation concentrations ($Ca^{2+}$, $Mg^{2+}$, $Na^+$) were analyzed on a Perkin Elmer Analyst 200 Atomic

Absorption Spectrometer at York University. A subset of collected samples were analyzed for total
dissolved Fe at the University of Alberta on an Inductively Coupled Plasma - Optical Emission
Spectrometer (Thermo Scientific ICAP6300), to allow for the correction of our Specific UV Absorbance
results (see below). DOC samples were analyzed on a Shimadzu TOC-V analyzer; DOC was calculated as
the mean of the best 3 of 5 injections with a coefficient of variance of <2%. A Picarro liquid water
isotope analyzer was used to measure stable water isotope samples at the University of Alberta,
following filtration (0.45 µm cellulose acetate, Sartorius)  into 2 mL autosampler vials (National
Scientific), without headspace. The precision of our $\delta^{18}O$ analysis is ± 0.2%. The radiocarbon signature of
DOC was measured following extraction and purification at the A.E. Lalonde AMS facility (University of
Toronto) using a 3MV tandem accelerator mass spectrometer (High Voltage Engineering) following
established methodologies (Lang et al., 2016; Palstra and Meijer, 2014; Zhou et al., 2015).

3.3.2 Total suspended sediments

Samples for total suspended sediments (TSS) were filtered in the field for later analysis, ensuring

that there was enough sediment on the pre-combusted (475$^o$C, 4 hours) and pre-weighed GF/F filters.
Filters were stored frozen, dried at 60$^o$C for 8 hours, placed in a desiccator overnight and promptly
weighed. TSS was calculated as the difference in filter weight before and after sediment loading, divided
by volume filtered.

3.3.3 Dissolved organic matter spectral characteristics

DOM composition was assessed using absorbance-based metrics. A 5 cm quartz cuvette was



used to obtain UV-visible spectra data from 250-750 nm, using a Genesys 10 UV-Vis spectrophotometer.
A baseline correction was applied to eliminate potential interference from particles following Green &
Blough (1994). Specific UV absorbance at 254 nm (SUVA$_{254}$), which is correlated with DOM aromaticity
(Weishaar and Aiken, 2003), was calculated by dividing the decadal absorbance at 254 nm (m$^{-1}$) by the
DOC concentration (mg L$^{-1}$). SUVA$_{254}$ values were corrected for Fe interference following Poulin et al.
(2014) using maximum Fe concentrations from laboratory analyses or as reported in Malone et al.
(2013). Spectral slopes between 275 and 295 nm, and 350 and 400 nm (S$_{275-295}$, S$_{350-400}$) were calculated
following Helms et al. (2008), and are reported as positive values to adhere to mathematical
conventions. Slope ratios (S$_R$), which correlate with DOM molecular weight (Helms et al., 2008), were
calculated as the ratio of S$_{275-295}$ to S$_{350-400}$.

*3.4 Statistical analyses*
Statistical analyses were completed in R version 3.1.3 (R Core Team, 2015) using packages 'nlme'
(Pinheiro et al., 2015), 'lmtest' (Zeileis and Hothorn, 2002), 'lmSupport' (Curtin, 2015), 'car' (Fox and
Weisberg, 2011), and 'zoo' (Zeileis and Grothendieck, 2005). The effect of slumping on stream chemistry
and optical characteristics was assessed using linear mixed effects models in the 'nlme' package of R. For
each parameter, analyses were split into two separate models that included data for upstream and
downstream chemistry, and upstream and within-slump chemistry. We used this approach to separately
assess the effects of slumping downstream of slump systems, and to compare the composition of slump
runoff to nearby, pristine environments. For each analysis, we included slump location (see Table 1) as a
random effect, and considered models that either nested Julian date within the random effect of slump
location, or allowed Julian date to occur as a fixed effect. The best model was chosen using AIC, and
best-fit models were refit with a variance structure to ensure that model assumptions were met. The
variance structures varIdent (for within-slump site and slump location) and varFixed (for Julian date)
were used together (using varComb) and in isolation for this purpose (Zuur et al., 2009). AIC values for



the weighted and un-weighted models were again compared to choose a final model of best fit for each
analysis.

We used the high-frequency data from slump FM3 to assess how environmental conditions

(rainfall, temperature, solar radiation) and TSS affect DOC delivery to slump-affected systems. To do
this, we conducted multiple linear regressions, using AIC values to determine models of best fit
(Burnham and Anderson, 2002). To enable a specific assessment of environmental controls on
downstream DOC flux, upstream DOC flux was separated out into a distinct regression analysis, because
upstream DOC flux was strongly correlated with flux downstream, and therefore overwhelmed all
environmental variables in the downstream model. Models were tested for serial correlation using the
auto-correlation function (ACF), and models with variance inflation factors greater than 10 or significant
Durbin Watson test results (indicative of correlated variables; Durbin & Watson, 1950; Hair et al., 1995)
were discarded. Residuals were examined to ensure the model was a good fit for the data (Zuur et al.,
2009). We considered both time-of-sampling (0 h) and past (48, 72, and 120 h) environmental conditions
in our analyses. Because cumulative values for environmental variables (i.e. accumulated rainfall in the
previous 48, 72 and 120 h) showed a strong positive correlation to one another, we used temporally
shifted data (i.e. rainfall 48, 72 and 120 h prior to the DOC flux measurement) in the final model. Similar
models were also constructed to examine the effects of environmental drivers on DOC concentration.
Finally, differences in paired upstream-downstream measures of DOC flux and concentration at slump
FM3 were assessed using a Wilcoxon Signed Rank Test, a non-parametric analog to the paired-t test.


4 Results
*4.1 DOC concentration across slump sites*

On the Peel Plateau, DOC concentrations consistently declined downstream of slumps, when

compared to paired, upstream locations (p<0.001; Fig. 2; Table 2). Although this effect was modest



(typically less than 20%; Fig. 2), it was consistent across all slump sites. In contrast, comparisons of
upstream and within-slump sites showed no consistent trend in DOC concentration, when evaluated
across all slump locations (p=0.153; Fig. 2; Table 2). Instead, the effects of slumping on the DOC
concentration of slump runoff appeared to vary by site. At the largest, most well-developed slump
complexes (FM4, FM2, and FM3), where debris tongues are extensive and thaw extends well into the
deepest layer of Pleistocene-aged glacigenic materials, DOC concentrations tended to be lower in slump
runoff than at the paired upstream sites (Fig. 2). At more modestly-sized slump sites (HB, HC, and HD),
where the modern and relict Holocene active layers form a greater proportion of the actively thawing
headwall, within-slump DOC concentrations tended to be higher than values upstream (Fig. 2). Within
each site, DOC concentrations were relatively consistent across the 2-3 sampling periods. However,
there was significant variation in DOC concentration between slump locations (i.e., across the Peel
Plateau landscape; Fig. 2).

*4.2 Bulk chemistry of pristine waters and slump runoff*

To better understand how the input of slump runoff affects downstream DOC, we examined

concentrations of conservative ions, conductivity and TSS as 'tracers' of slump activity, because these
constituents have previously been shown to be significantly affected by slumping in this region (Kokelj et
al., 2005, 2013; Malone et al., 2013; Thompson et al., 2008). Major ion ($Ca^{2+}$, $Mg^{2+}$, $Na^+$) concentrations
in slump runoff were considerably greater than in pristine streams (a 2.7 to 11.7-fold increase; Fig. 3b-d;
Table 2). These patterns were similar, though muted, at slump-affected downstream sites, where major
ion concentrations were 1.5 to 3.5-fold greater than at pristine sites (Fig. 3b-d; Table 2). Average
conductivity also increased significantly as a result of slumping (p< 0.001; Table 2): within-slump sites
had conductivity values that were 9.2-fold greater than upstream sites, while downstream values were
2.6 times greater than those upstream (Fig. 3e). Finally, TSS was also significantly elevated at slump-
affected sites (p< 0.001; Table 2) with levels being more than two orders of magnitude greater within



slumps when compared to upstream, and more than one order of magnitude greater downstream,
when compared to upstream sites (Fig. 3a).

*4.3 Spectral and isotopic characteristics*
SUVA$_{254}$, which is positively correlated with DOM aromaticity (Weishaar and Aiken, 2003), was
significantly lower within slumps, and downstream of slumps, than in upstream, pristine, environments
(p<0.001; Fig. 4; Table 2). Average within-slump SUVA$_{254}$ was less than half of that observed for pristine
waters (Fig. 4), while downstream values declined by approximately 20%. In accordance with the
SUVA$_{254}$ results, S$_{275-295}$, S$_{350-400}$, and S$_R$ were all significantly greater within slumps when compared to
upstream sites (p<0.001; Fig. 4; Table 2), indicating lower DOM molecular weight within slumps (Helms
et al., 2008). Differences in slope parameters between upstream and downstream locations were muted
relative to the within-slump: upstream comparisons (Fig. 4), with S$_{275-295}$ (p=0.011) and S$_R$ (p<0.001)
increasing significantly, but more modestly, downstream of slumps, and S$_{350-400}$ declining slightly
(p=0.001; Fig. 4; Table 2).
Upstream $\delta^{18}$O averaged -20.1‰ ± 0.12, which corresponds to a modern $\delta^{18}$O signature for this
region (Lacelle et al., 2013; Fig. 5). Within-slump $\delta^{18}$O was discernibly depleted when compared to
upstream locations, with average values of -22.7‰ ± 0.72, which falls between previously-identified
regional endmembers for Pleistocene-aged ground ice (18,100 ± 60 $^{14}$Cyr BP) and the modern active
layer (Lacelle et al., 2013; Fig. 5). Within-slump $\delta^{18}$O was also much more variable between RTS features
than upstream and downstream $\delta^{18}$O values. Similar to upstream sites, downstream $\delta^{18}$O clustered near
the modern active layer $\delta^{18}$O endmember, but with a small depletion that was consistent with a
contribution from slump inflow (-20.7‰ ± 0.21).
To further investigate the effect of water source on DOM composition, we examined the
relationship between SUVA$_{254}$ and $\delta^{18}$O. More depleted samples taken from within-slump sites had
clearly depressed SUVA$_{254}$ values when compared to samples with more enriched $\delta^{18}$O (Fig. 5). Of the



large, most well-developed slumps that were identified in Section 4.1, two (FM2 and FM3), in addition
to site HB, had $\delta^{18}$O values that were more depleted than the Holocene-aged icy diamicton values
reported in Lacelle et al. (2013), suggesting some contribution of runoff from older, Pleistocene-aged
permafrost (Fig. 5). It is likely that the $\delta^{18}$O signal at the relatively stable mega-slump site (FM4) was
somewhat diluted by the 7.2 mm of rainfall that fell in the 48 hours preceding our sample. Although
sites FM3 and SD received 12.4 and 3.5 mm of rain, respectively, in the 48 hours prior to sampling, these
are both much more active slump sites, and thus less prone to dilution of the slump outwash signature.
There was no significant rainfall immediately preceding sampling at any other sites.

The radiocarbon signature of DOC from upstream and within-slump locations at sites FM4, FM2,

FM3, and SD largely mirrors the $\delta^{18}$O results. DOC from sites upstream of slump disturbances was
approximately modern in origin (ranging from 217 ± 24 $^{14}$C yr BP to modern in age; Table 3). In contrast,
within-slump waters from site FM2 and FM3 were early Holocene-aged (9592 ± 64, and 8167 ± 39 $^{14}$C yr
BP, respectively; Table 3). Slump runoff from site SD was older than at upstream sites, but younger than
for the larger slumps, described above (1157 ± 23 $^{14}$C yr BP; Table 3).

*4.4 Patterns and environmental drivers of DOC flux*

Similar to our findings for the distributed sampling scheme, downstream DOC concentration was

consistently lower than concentrations upstream, across the 19 paired measurements taken at the
intensively studied slump site (slump FM3; p<0.001, N=19, W=0; Wilcoxon Signed Rank Test). To explore
environmental drivers of DOC movement within this landscape, however, we focus on DOC flux (as mg s$^-$
$^1$), which allows a direct assessment of slump-mediated DOC addition to this system. Downstream DOC
flux (mg s$^{-1}$) tended to be slightly greater than upstream flux on most, but not all, sampling occasions
(Fig. 6). As a result, paired comparisons indicate no statistical difference between upstream and
downstream DOC flux at this site (Wilcoxon signed rank test; p=0.096, N=19, W=53). Because upstream
and downstream DOC flux were strongly correlated to one another (r$^2$ = 0.94; p<0.0001), our



downstream model was run without upstream DOC flux as a predictor variable. The best-fit multiple
linear regression model for downstream DOC flux ($r^2$ = 0.84; p<0.01) retained seven variables, of which
two were significant (Table 4). Of these, air temperature (72h prior to sampling) showed a negative
relationship with downstream DOC flux and rainfall (0h; time of sampling) showed a strong positive
relationship (Table 4). The best-fit model for upstream DOC flux ($r^2$ = 0.87; p<0.001) also retained seven
variables, of which four were significant (p<0.05; Table 4). Similar to the downstream analysis, air
temperature (0h, 72h) had a negative relationship, and time-of-sampling (0h) rainfall had a strong
positive relationship, with DOC flux (Table 4). However, 120h rainfall showed a negative relationship
with DOC flux in this model. Regressions exploring controls on downstream DOC flux relative to
upstream flux (i.e., as a ratio, or the difference between the two values) were not significant. Models
exploring controls on upstream and downstream DOC concentration were also relatively similar to one
another, showing strong, positive relationships between DOC concentration and air temperature, and
more modest negative relationships between DOC concentration and net radiation (Table 4).


**5. Discussion**
*5.1 Retrogressive thaw slumps and carbon delivery to streams of the Peel Plateau*

In Eastern Siberia (Drake et al., 2015; Mann et al., 2015; Vonk et al., 2013b), Alaska (Balcarczyk

et al., 2009; Abbott et al., 2014), and the Canadian High Arctic (Melville Island; Woods et al.,
2011),permafrost slumping has been associated with significant increases in DOC mobilization to
streams. Our data show that this was not the case on the Peel Plateau, where the landscape-induced
variation in DOC concentration among pristine stream sites was much greater than the change in stream
water DOC as a result of slumping. Across all of our study sites, DOC concentrations consistently
decreased downstream of slumps when compared to upstream locations. In contrast, comparisons of
channelized slump runoff (our within-slump sites) and paired un-affected sites showed no consistent





DOC trend. Instead, DOC concentrations in slump runoff were either greater than, or less than, their
comparison upstream locations, in a manner that differed depending on slump morphological
characteristics such as slump size and headwall height (*Fig. 1; see further discussion in Section 5.3*). The
moderate effect of slumping on DOC concentration occurred despite the significant influence of these
disturbances on the delivery of many biogeochemical constituents to recipient streams. For example,
conductivity was approximately one order of magnitude greater, and TSS two orders of magnitude
greater, in slump-derived runoff than at upstream, un-affected sites.

Decreasing DOC concentrations downstream of slumps, despite increasing concentrations of

indicators of slump activity (major ions, TSS) could have several, potentially co-occurring causes. In some
locations, decreases may be partially caused by low DOC concentrations in slump outflow (a dilution
effect; see slumps FM2, FM3, and FM4; Fig. 2). However, field evidence suggests that DOC sorption to
suspended inorganic sediments could also play a role in regulating DOC dynamics in slump-affected
systems on the Peel Plateau. At multiple sites (HB, HC, and HD), DOC concentrations declined
downstream of slumps despite a modest elevation in DOC concentration in slump drainage waters (Fig.
2). Thermokarst contributes significant amounts of glacigenic sediment to fluvial systems on the Peel
Plateau (Kokelj et al., 2013), and this material is fine-grained (e.g., sediments from the FM3 headwall
have been classified as silty clay; Lacelle et al., 2013). DOC sorption can occur in seconds to minutes in
freshwater systems (Qualls and Haines, 1992), with fine-grained materials being particularly conducive
to these processes (Kothawala et al., 2009). Data from site FM3, where we have upstream and
downstream discharge data coupled with DOC and TSS concentrations at upstream, downstream, and
within-slump locations on two separate dates, allows us to assess possible DOC sorption at this site. On
these dates, DOC flux declines downstream of the slump (i.e., $\text{flux}_{DOCdown} < \text{flux}_{DOCup}$), despite a clear and
measurable efflux of DOC from within the slump (calculated as $[DOC]_{within} \bullet (\text{discharge}_{down} - \text{discharge}_{up})$;
Fig. 7). This same calculation using TSS as a conservative tracer of slump activity shows the calculated
within-slump flux of TSS (as $[TSS]_{within} \bullet (\text{discharge}_{down} - \text{discharge}_{up})$) to be almost identical to the



difference in TSS flux between downstream and upstream locations (as flux$_{TSSdown}$ – flux$_{TSSup}$; Fig. 7). Thus,
it is likely that relatively rapid processes such as sorption are affecting DOC dynamics downstream of
slumps on the Peel Plateau.

The decrease in DOC concentration downstream of Peel Plateau slumps is similar to, but more

muted than results for lakes in this region, where following slump stabilization, lakes are characterized
by increases in conductivity, clear decreases in DOC concentration, and a strong negative correlation
between these two parameters. The greater magnitude of effect for lakes in this region is likely caused
by substantial particle settling in lentic environments, which enables DOC scavenging with the inorganic
sediment inputs of thermokarst (Kokelj et al., 2005). Although decreasing DOC with RTS activity on the
Peel Plateau contrasts with work to-date in other regions (e.g., Abbott et al., 2014; Vonk et al., 2013a),
ice-marginal glaciated landscapes intensely affected by RTS are common throughout the western
Canadian Arctic, and many other Arctic regions (Kokelj et al., 2017b). In general, this terrain type is
typically characterized by thick, mineral-rich but carbon poor tills, which with their high ice contents are
predisposed to climate-driven thaw slumping and release of glacigenic sediments. Thus, it seems likely
that the processes we observe are not limited to the Peel Plateau: research to quantify DOC
'sequestration' via sorption processes seems warranted across regions where thermokarst intensifies
the transport of mineral-rich sediments to downslope aquatic systems.

*5.2 The effect of retrogressive thaw slumps on DOM composition*

Despite the fact that DOC concentrations did not increase in RTS-affected streams, SUVA$_{254}$ and

absorbance metrics clearly indicate that slump-derived DOM on the Peel Plateau is compositionally
different than DOM from upstream locations. Upstream waters had significantly higher SUVA$_{254}$ values
compared to downstream and within-slump sites (Table 2, Fig. 4). Similarly, while the average S$_R$ of Peel
Plateau upstream waters (0.74 ± 0.005) was within the range of S$_R$ typically associated with fresh,
terrestrial DOM (~ 0.70; Helms et al., 2008), values were significantly greater within-slump (0.92 ± 0.015)





and downstream (0.89 ± 0.009) (Table 2, Fig. 4), indicating decreasing DOM molecular weight as a result
of RTS activity. High SUVA$_{254}$ values accompanied by low S$_R$ at upstream sites suggest that water flow in
undisturbed catchments is restricted to shallow, organic-rich flowpaths through the active layer, with
permafrost inhibiting water contributions from deeper, groundwater or mineral-associated sources
(Balcarczyk et al., 2009; MacLean et al., 1999; Mann et al., 2012; O'Donnell et al., 2010). In contrast,
within-slump and downstream measurements indicate a clear transition in DOM source.

The comparatively low SUVA$_{254}$, and high S$_R$ values for downstream and within-slump sites

indicate that permafrost-derived carbon on the Peel Plateau is similar in its composition to permafrost
carbon from other regions. For example, SUVA$_{254}$ values were low in waters draining active thaw slumps
when compared to stabilized and undisturbed sites on the North Slope of Alaska (Abbott et al., 2014),
while in Siberia, [14]C-depleted DOM from small tributary streams affected by thermokarst had lower
SUVA$_{254}$ values compared to younger DOM from the Kolyma River mainstem (Mann et al., 2015; Neff et
al., 2006). Although SUVA$_{254}$ values for waters draining Peel Plateau thaw slumps are slightly lower than
those reported for Siberian Yedoma disturbances (Mann et al., 2015), the overall similarity of
permafrost-derived DOM composition across these various regions is striking, given the regional
differences in permafrost origin and depositional history. For example, while the DOM released by
permafrost thaw on the Peel Plateau is till-associated, and early-Holocene in mean age, east Siberian
Yedoma is composed of loess-derived Pleistocene deposits that sequestered carbon in association with
synengetic aggradation of permafrost. This suggests that common processes may enable the organic
matter contained in permafrost soils to become compositionally similar across diverse Arctic regions.
Such compositional similarity also indicates that permafrost-origin DOM from the Peel Plateau – similar
to that from other regions (Abbott et al., 2014; Drake et al., 2015) – may be readily degraded by
bacteria, despite the divergent origin of this carbon.



*5.3 The effect of slump morphometry on runoff water biogeochemistry*
$\delta^{18}$O and DO$^{14}$C data provide further evidence that intense slumping enables novel sources of
water and solutes to be transported to fluvial systems on the Peel Plateau. For most of the RTS features
that we studied, the $\delta^{18}$O signature of within-slump waters ranged from those similar to the 'icy
diamicton' that overlies the early Holocene thaw unconformity, to those for underlying Pleistocene-aged
ground ice (Lacelle et al., 2013; Fig. 5). Similarly, DO$^{14}$C from a subset of sites indicates slump-derived
DOC is early Holocene-aged for all but the shallowest slump surveyed. This suggests that our slump
outflow samples were likely comprised of a mixture of Pleistocene-, Holocene-, and modern-sourced
water (see Fig. 1c-e), but that the contribution of these end-members varied across slumps.
The between-site variation in $\delta^{18}$O signature (Fig. 5) and relative DOC concentration (Fig. 2b) of
slump runoff waters appears to be related to differences in slump morphometry (size, headwall height,
and the length and area of the debris tongue; *see Table 1 and Fig. 1c-e*) across sites. The well-developed,
larger slump complexes (FM4, FM2 and FM3) were more likely to have $\delta^{18}$O signatures that lie between
end-member values for icy diamicton and Pleistocene-aged ground ice (Fig. 5; although note that dry
and stabilized FM4 differs somewhat from this trend). These well-developed slumps also stood out as
displaying within-slump DOC concentrations that were lower than at upstream comparison sites (Fig.
2b). The headwall exposure at these largest slumps exposes Pleistocene-aged permafrost to several
metres depth (see Fig. 1c), while the evacuation of scar zone materials have produced extensive debris
tongues up to several kilometers long (Table 1, Figs. 1b, S1e and S1g). This significant exposure of
mineral-rich, Pleistocene-aged glacial till contributes solutes from low-carbon mineral soils to runoff,
while entraining fine-grained sediments which provide mineral surface area for possible DOC
adsorption. Adsorption may be further enhanced as slump and stream runoff continue to entrain
sediments as flows incise the lengthy debris tongue deposits. In contrast, slumps with slightly shallower
headwalls (HA, HB, HC, HD; see Fig. 1d), and less well-developed debris tongues (Table 1), appear to
elicit a slightly different response than the largest slumps discussed above. At these mid-sized sites,



within-slump DOC concentrations were typically higher than those found at upstream comparison sites
(Fig. 2b), which may reflect the greater relative inputs from thawing of the Holocene-aged active layer,
and decreased interaction with debris tongue deposits at these smaller disturbances. Similarly, runoff
$\delta^{18}$O tends to lie between Holocene and modern end-member values at these sites (though note the
more depleted value for HB; Fig. 5), indicating a lower relative contribution of Pleistocene-aged ground
ice.

Finally, the youngest and shallowest slump surveyed (SD), exposes only near-surface permafrost

soils for leaching and geochemical transport (Figs. 1e and S1; Table 1), and not the underlying mineral
and ice-rich glacigenic substrates. Accordingly, the effects of slumping on stream chemistry, optical
parameters, and isotopes appear muted at SD when compared to the larger slumps discussed above.
These morphometry-related shifts in downstream effect suggest that we should expect non-linearity in
the biogeochemical response as slump features develop over time, particularly if slumping continues to
intensify with future warming on the Peel Plateau (e.g., Kokelj et al., 2017b). Long-term monitoring, and
the incorporation of non-linearity into models predicting future change, are clearly warranted for the
Peel Plateau and elsewhere in the Arctic.

*5.4 Environmental controls on DOC flux and concentration*

Air temperature and rainfall exerted the strongest control on DOC flux at our intensively studied

site (slump FM3; Fig. 6; Table 4). Upstream of the slump, rainfall was positively correlated, and air
temperature negatively correlated, with DOC flux. However, precipitation events are negatively related
to temperature at this site (Fig. 6), suggesting that at the single-season scale of our investigation,
precipitation served as the primary environmental control on DOC flux. DOC concentration was
relatively constant with discharge at the upstream site (r=-0.342, p=0.151), indicating that precipitation
controls DOC flux largely as a result of changes in water flow at this site, and that DOC was not source
limited over the time scale of our investigation. DOC concentration was positively related to





temperature, however (Table 4), suggesting that biological activity is an important regulator of within-
soil DOC production (c.f. Pumpanen et al., 2014). These upstream-of-slump results are consistent with
work from other undisturbed permafrost and boreal regions, where precipitation and catchment runoff
have been shown to control DOC flux in streams (Prokushkin et al., 2005; Pumpanen et al., 2014), and
increasing temperature has been shown to increase DOC production in soils (Christ and David, 1996;
Neff and Hooper, 2002; Prokushkin et al., 2005; Yanagihara et al., 2000). They are also consistent with
the concept that the impermeable permafrost barrier forces precipitation to travel through the shallow
active layer, where high hydraulic conductivity leads to rapid transport of carbon into fluvial systems,
and little degradation in soils ( O'Donnell et al., 2010; Striegl et al., 2005).

Slumping did not significantly modify downstream DOC flux at the intensively studied slump site,

when compared to DOC flux upstream of slump FM3 (Fig. 6; Section 4.4). Although concentration
consistently declined downstream at this site (Sections 4.1 and 4.4), downstream flux was either slightly
higher, or slightly lower, than upstream values, across the multiple measurement points that we
considered. Concordant with this finding, neither the ratio of (downstream: upstream) or difference
between (downstream – upstream) upstream and downstream DOC flux could be explained by any of
our environmental variables, while downstream flux showed an almost identical relationship with
environmental controls as those upstream (Table 4). The lack of clear environmental control on relative
downstream: upstream DOC flux occurred despite the fact that precipitation has been shown to be a
strong driver of ablation and sediment movement from slump features on the Peel Plateau, at time
scales similar to those used for this work (Kokelj et al., 2015).

Considering the Peel Plateau landscape as a whole, it appears that precipitation serves as a

primary, positive control on DOC flux. Thus, this study adds DOC production to the list of changes – such
as increasing slump activity and sediment mobilization – that can be expected with the increases in
precipitation that are underway in this region, and are expected throughout the Arctic (Kokelj et al.,
2015; Walsh et al., 2011). However, it appears that slumping does not over-ride the landscape-scale



control on DOC flux in this system – at least at the scale of this single-season – perhaps because
processes like DOC sorption mask the influx of slump-derived DOC (Fig. 6). This result clearly highlights
the complexity of the interaction between changing climatic parameters and DOC dynamics on the Peel
Plateau, where slump features of increasing size incorporate thawing till, glaciolacustrine, glaciofluvial,
and organic deposits; additionally drain contemporary active layers across a shrub-tundra to spruce
forest upland gradient; and where DOC dynamics are variably affected by both water and carbon
generation across these landform types, and biogeochemical interactions such as mineral adsorption in
recipient systems. It also underscores the need for future work to tease apart the interactions between
changing climatic parameters, slump development, and resultant biogeochemical effects; both on the
Peel Plateau and across the Arctic, where environmental controls on slump activity and thus
downstream biogeochemistry can be expected to show marked regional variation (see for example,
work from Eureka Sound; Grom & Pollard 2008).

*5.5 Study implications and future research directions: dissolved carbon mobilization across diverse*
*permafrost landscapes*

Carbon dynamics in Arctic aquatic systems are influenced by numerous factors, including

geology, Quaternary and glacial history, soil composition, vegetation, active layer dynamics and the
nature and intensity of thermokarst. As a result, the effect of permafrost thaw on DOC concentration
and flux should – at a fundamental level – vary across broad, regional scales. This study demonstrates
that we can expect large inter-regional variations in DOC transport to streams in response to permafrost
degradation, and that results from multiple regions are needed to understand change across the Arctic
as whole. The declines in DOC concentration downstream of slumps on the Peel Plateau differ strikingly
from what has been found in eastern Siberia and regions of Alaska, for example, where thermokarst
releases substantial quantities of DOC (e.g., Spencer et al. 2015), and significantly increases DOC
concentrations in downstream systems (Abbott et al. 2015). Modelling efforts that incorporate



information concerning the geology and Quaternary history of landscapes that are being thawed, the
physical and geochemical composition of permafrost soils, and the nature and intensity of the
thermokarst processes within landscapes would clearly enable more accurate predictions of how carbon
delivery from land to water will respond to climate change on a pan-Arctic scale.

At finer scales, however, this work underlines the variability of thermokarst effects within

regions, and the local-scale control on this variability. On the Peel Plateau, for example, between-site
difference in the biogeochemical effect of thermokarst corresponds to variation in soil stratigraphy (i.e.,
the relative depth of the paleo-active layer) and ever-evolving slump morphometry. Although striking
within-region variability in the biogeochemical effects of thermokarst has been seen elsewhere (e.g.,
Watanabe et al., 2011), it occurs a result of very different landscape-level drivers. This landscape-
specificity also extends to the non-linearity of the biogeochemical response as slump features develop
over time. The changing response of downstream biogeochemistry with slump development is very
different on the Peel Plateau, for example, than in other regions (e.g., Abbot et al. 2015), while non-
linearity can also be expected to extend to different types of permafrost thaw, such as increasing active
layer thickness (Kokelj et al. 2002, Vonk et al. 2016). Only with a tiered approach, where we focus within
regions to understand local controls and changing effects over time, and across regions to document
how predictable, broad-scale variation affects the nature of thermokarst effects, will we be able to truly
understand the future biogeochemical functioning of thermokarst-affected landscapes at the pan-Arctic
scale.


**Acknowledgements**
Financial support for this research was provided by Ontario Graduate Scholarship, York University
Fieldwork Cost Fund, York University Research Cost Fund, Northern Scientific Training Program, NSERC
Discovery and Northern Research Supplement grants to SET, the Campus Alberta Innovates Program,



and the Polar Continental Shelf Program.  We would like to thank Scott Zolkos for his support as a field
assistant and for the production of Figure 1; S. Tetlichi, D. Neyando, and P. Snowshoe for field sampling
assistance; and the Tetlit Gwich'in (Fort McPherson) Renewable Resources Council.  Sarah Shakil and
Scott Zolkos assisted with the collection of samples for DO$^{14}$C; Justin Kokoszka performed geospatial
calculations of slump area and debris tongue length.





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

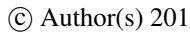



**Table 1:** Slump characteristics and sampling information for eight retrogressive thaw slumps sampled
during the 2014 field season on the Peel Plateau, NWT, Canada. Characteristics are derived from
published values and field estimations.

| Slump location | Sample dates (Julian day)[a] | Latitude | Longitude | Scar zone (ha) | Debris tongue (m)[b] | Headwall height (m) |
|---|---|---|---|---|---|---|
| FM4 | 202, 210, 223 | 67 16.679 | -135 09.573 | 8.8 | 960 | 16 to 20[d] |
| FM2 | 200, 209, 222 | 67 15.462 | -135 14.216 | 31.7 | 1529 | 25[e] |
| FM3 | 197, 212 | 67 15.100 | -135 16.270 | 6.1 | 576 | 10[e] |
| SD | 196, 213, 234 | 67 10.818 | -135 43.630 | 3.3 | NA | 2 – 4[d] |
| HA | 190, 229 | 67 09.057 | -135 41.121 | 5.9 | 288 | 6 – 10[d] |
| HB | 190, 229 | 67 14.397 | -135 49.167 | 13.6[c] | 257 | 6 – 10[d] |
| HC | 190, 229 | 67 19.652 | -135 53.620 | 10.3, 10.3[c] | 408 | 6 – 10[d] |
| HD | 190, 229 | 67 24.025 | -135 20.048 | 1.8 | 137 | 6 – 10[d] |
| Weather Station | | 67 14.756 | -135 12.920 | | | |


[a] Excludes samples for the FM3 'environmental controls' analysis which was conducted using samples
from 17 additional dates; HD, Julian date 229 did not include a within-slump sample
[b] The length of debris tongue measured from the base of the debris scar, along the valley bottom stream
[c] Site HB is comprised of two smaller slump features that have merged into the scar zone delineated
here; site HC is comprised of 5 separate slump features that have merged into the two scar zones
delineated here
[d] Rough estimates by field crews over 2014 and 2015 field seasons
[e] (Kokelj et al., 2015)







**Table 2:** Results of the mixed-effects models used to assess the effects of slumping on stream water
chemistry and optical characteristics. Downstream models incorporated data from downstream and
upstream sites; within-slump models incorporated data from within-slump and upstream sites. Further
details on the statistical approach are provided in Section 3.4.

|  | Downstream | | | Within-slump | | |
|---|---|---|---|---|---|---|
|  | df | t | p | df | t | p |
| DOC | 20 | -12.895 | <.0001 | 30 | -1.468 | 0.153 |
| Na | 33 | 9.662 | <.0001 | 30 | 7.278 | 0.000 |
| Ca | 33 | 9.767 | <.0001 | 30 | 4.782 | 0.000 |
| Mg | 33 | 6.166 | <.0001 | 30 | 8.593 | 0.000 |
| Conductivity | 32 | 43.083 | <.0001 | 30 | 11.895 | 0.000 |
| TSS | 29 | 6.692 | <.0001 | 28 | 2.187 | 0.037 |
| SUVA | 31 | -5.296 | <.0001 | 30 | -35.052 | 0.000 |
| $S_R$ | 31 | 5.092 | <.0001 | 31 | 8.065 | 0.000 |
| $S_{275}$ | 30 | 2.695 | 0.011 | 31 | 8.159 | 0.000 |
| $S_{350}$ | 31 | -3.595 | 0.001 | 31 | 16.665 | 0.000 |







**Table 3:** Measured fraction modern (F$^{14}$C) and estimated calendar years before present for $^{14}$C of
dissolved organic carbon samples collected upstream of, and within drainage waters of, selected slump
sites. Data were collected during the summer of 2016. nc indicates sample not collected.

| Site | F$^{14}$C | | $^{14}$C yr BP | |
|---|---|---|---|---|
| | Upstream | Within-slump | Upstream | Within-slump |
| FM4 | 0.9734 ± 0.0029 | nc | 217 ± 24 | nc |
| FM2 | 0.9764 ± 0.0032 | 0.3030 ± 0.0024 | 192 ± 27 | 9592 ± 64 |
| FM3 | 1.0023 ± 0.0030 | 0.3618 ± 0.0018 | modern | 8167 ± 39 |
| SD | 1.0216 ± 0.0035 | 0.8659 ± 0.0025 | modern | 1157 ± 23 |



**Table 4:** Results of multiple linear regression analyses to assess environmental controls on upstream and downstream DOC flux, and upstream and downstream DOC concentration. nr indicates variables that were not retained in the best fit regression model; NA indicates variables that were not run in individual analyses. Significant p-values are indicated in bold text; marginal results (0.05 < p < 0.10) are indicated in italics. Model statistics are as follows: downstream flux $r^2$=0.84, $F_{7,11}$=8.25, p = 0.001; upstream flux $r^2$=0.87, $F_{7,11}$=10.79, p <0.001; downstream concentration $r^2$=0.85, $F_{4,14}$=19.57, p < 0.001; upstream concentration $r^2$=0.91, $F_{5,13}$=27.05, p < 0.001.

| Coefficient | Downstream DOC flux | | | Upstream DOC flux | | | Downstream DOC concentration | | | Upstream DOC concentration | | |
|---|---|---|---|---|---|---|---|---|---|---|---|---|
| | Estimate | t | p | Estimate | t | p | Estimate | t | p | Estimate | t | p |
| **Average Air Temperature (°C)** | | | | | | | | | | | | |
| 0 h | -67.08 | -1.685 | 0.120 | -115.96 | -3.286 | **0.007** | nr | nr | nr | 0.165 | 2.349 | **0.035** |
| 48 h | nr | nr | nr | 56.32 | 1.534 | 0.153 | **0.332** | 6.886 | **<0.001** | 0.396 | 5.510 | **<0.001** |
| 72 h | **-95.15** | **-2.594** | **0.025** | **-94.17** | **-2.717** | **0.020** | nr | nr | nr | nr | nr | nr |
| 120 h | nr | nr | nr | nr | nr | nr | 0.134 | 3.527 | **0.003** | 0.203 | 4.411 | **<0.001** |
| **Rainfall (mm)** | | | | | | | | | | | | |
| 0h | **116.13** | **5.411** | **<0.001** | 105.47 | 6.039 | **<0.001** | -0.066 | -1.967 | *0.069* | nr | nr | nr |
| 48h | nr | nr | nr | nr | nr | nr | nr | nr | nr | nr | nr | nr |
| 72h | nr | nr | nr | nr | nr | nr | nr | nr | nr | nr | nr | nr |
| 120h | -23.94 | -1.970 | *0.075* | -24.15 | -2.529 | **0.028** | nr | nr | nr | nr | nr | nr |
| **Average net radiation (W m⁻²)** | | | | | | | | | | | | |
| 0h | 4.96 | 1.286 | 0.225 | nr | nr | nr | -0.021 | -4.043 | **0.001** | -0.021 | -3.387 | **0.005** |
| 48h | nr | nr | nr | nr | nr | nr | nr | nr | nr | nr | nr | nr |
| 72h | 5.58 | 1.545 | 0.151 | 4.04 | 1.563 | 0.146 | nr | nr | nr | nr | nr | nr |
| 120h | nr | nr | nr | nr | nr | nr | nr | nr | nr | nr | nr | nr |
| **Total suspended sediment (mg L⁻¹)** | | | | | | | | | | | | |
| Downstream | *-0.02* | *-2.102* | *0.059* | NA | NA | NA | nr | nr | nr | NA | NA | NA |
| Upstream | NA | NA | NA | -0.32 | -1.626 | 0.132 | NA | NA | NA | -0.0006 | -1.627 | 0.128 |



**Figure captions:**

**Fig. 1***:* Panel A depicts the stream networks and location of the eight retrogressive thaw slumps studied on the Peel Plateau, Northwest Territories, Canada. Panel B depicts representative sampling locations at each slump site; FM3 depicted. Panels C-E depict representative thaw-slump headwall stratigraphies. Panel C shows a mega-slump (FM3, the smallest mega-slump, is depicted); panel D shows a moderate-sized slump (HB); panel E shows the smallest slump that was sampled (SD). In panels C and D, the approximate location of the modern active layer (a), early Holocene-aged relict active layer (b), and Pleistocene-aged glacigenic materials (c) is shown.

**Fig. 2:** The effect of retrogressive thaw slumps on stream water dissolved organic carbon (DOC) concentration. Each data point represents the mean and standard error of measurements across all sampling dates, as described in Table 1. The bottom two panels show the ratio of within-slump: upstream, and downstream: upstream DOC concentrations within individual slumps, with points indicating the mean and standard error of this ratio across sample dates.

**Fig. 3:** Box and whisker plots to illustrate the effects of retrogressive thaw slump activity on stream geochemistry. Each boxplot includes data from across all slumps and sampling periods, and indicates median values, $25^{th}$ and $75^{th}$ percentiles (box extremities), $10^{th}$ and $90^{th}$ percentiles (whiskers), and outlier points. U=upstream sites; W=within-slump sites; D=downstream sites.

**Fig. 4:** The effect of retrogressive thaw slumps on the optical properties of stream water dissolved organic matter. Each data point represents the mean and standard error of measurements across all sampling dates, as described in Table 1.

**Fig. 5:** Paired data on the oxygen isotopic composition of water ($\delta^{18}O$ ‰) and $SUVA_{254}$ (L mg $C^{-1}m^{-1}$) characteristics of dissolved organic matter (DOM), to demonstrate the relationship between source water age and DOM composition. $\delta^{18}O$ values for the modern active layer, icy diamicton, and Pleistocene-aged ground ice are from Lacelle et al. (2013): the modern active layer is equivalent ot the meteoric water line, icy diamicton has been aged to be Holocene era, and the value for Pleistocene-aged ground ice is the most enriched Pleistocene-aged value for this region.

**Fig. 6**: Environmental conditions (solar radiation, precipitation and mean daily air temperature) and DOC flux upstream and downstream of slump FM3 across a month-long sample period (July 12-August 12, 2014). Corresponding multiple linear regressions are described in Table 4.

**Fig. 7***:* Within-slump fluxes of dissolved organic carbon (DOC), and TSS, compared to the calculated (downstream - upstream) fluxes for these two constituents. TSS – a conservative tracer over short distances – shows an additive response where the measured within-slump flux is equivalent to the calculated (downstream - upstream) flux. In contrast, DOC shows clear evidence of downstream loss.






**Figure 1**




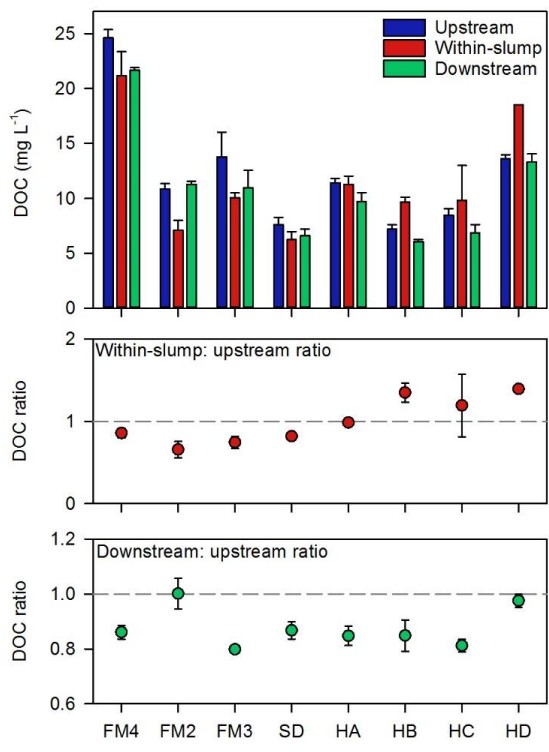


**Figure 2**





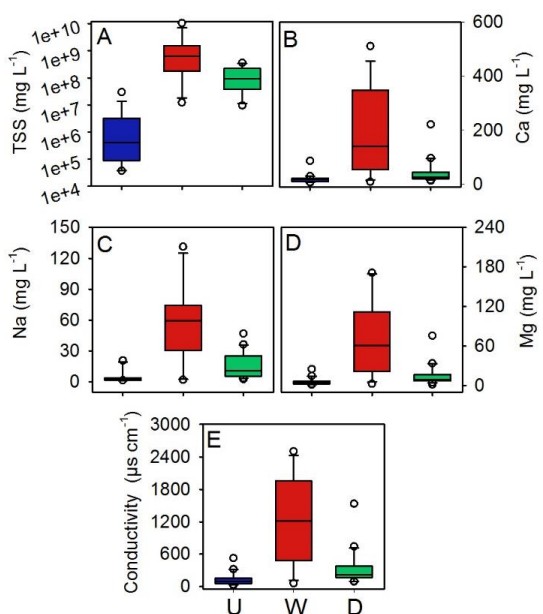


**Figure 3**





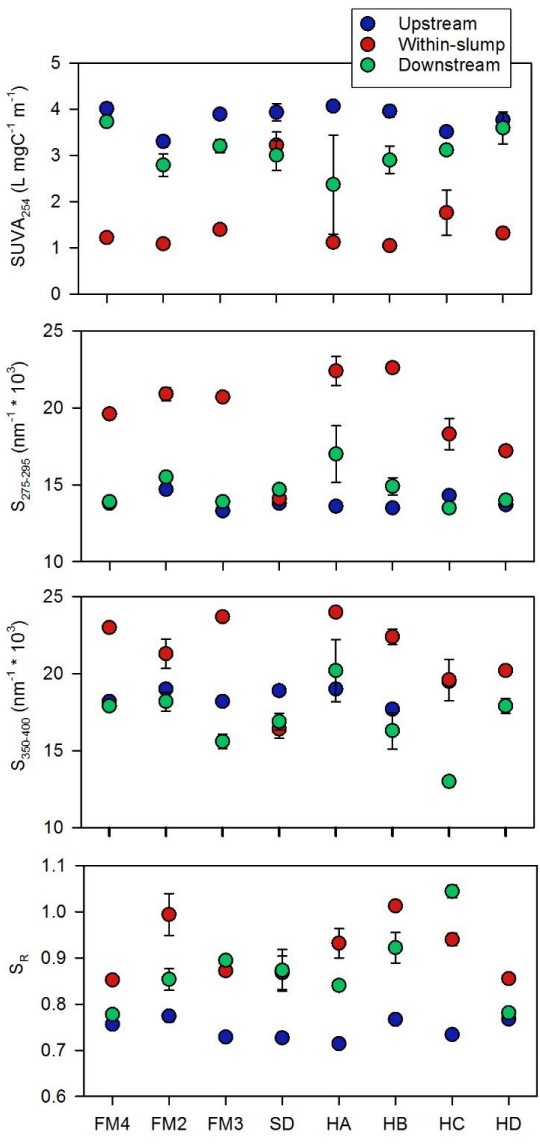

**Figure 4**



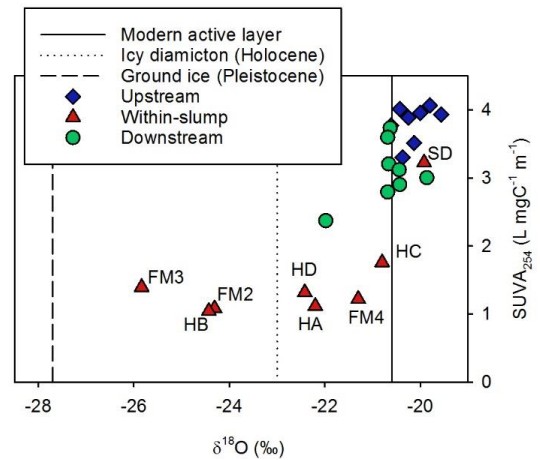


**Figure 5**






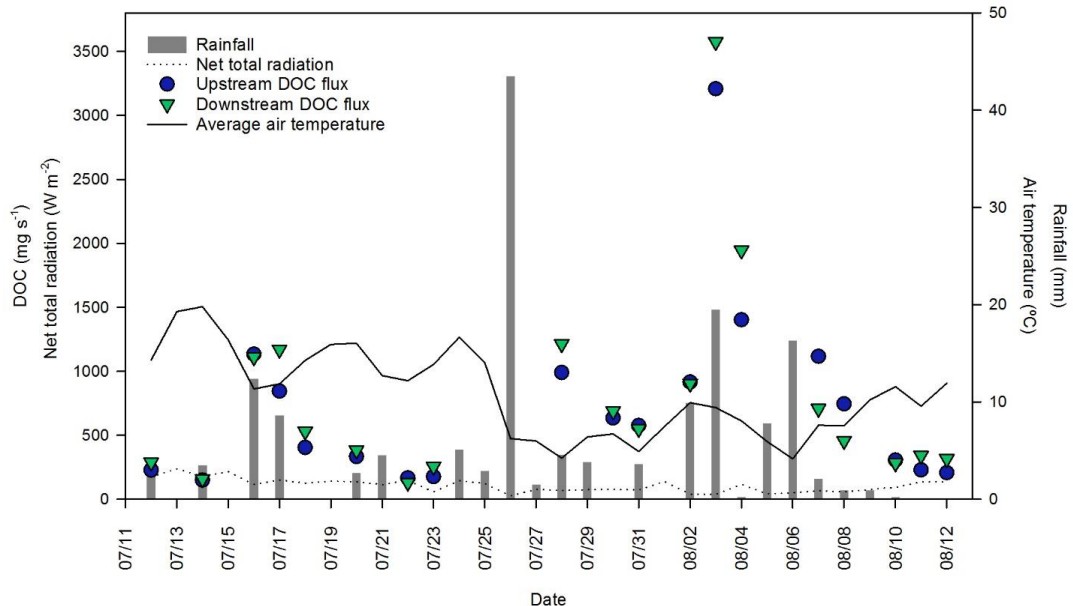


**Figure 6**



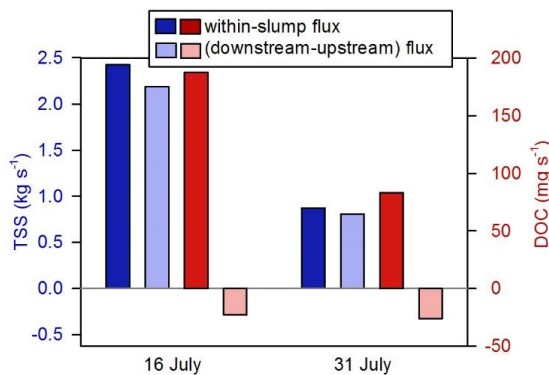



**Figure 7**