# Peer review of "Retrogressive thaw slumps temper dissolved organic carbon delivery to streams of the Peel Plateau, NWT, Canada"

_Biogeosciences, 2017_

## Referee Comment (RC1) · Anonymous Referee #1 · 25 Jul 2017

This manuscript represents an important step in our understanding of permafrost thaw dynamics from the Canadian Arctic. As the authors point out, the permafrost regions of the Arctic are not all similar, and region-specific work such as this are critical to our understanding of the impact of climate change on the Arctic as a whole. The key finding of the manuscript, that permafrost thaw slumps may in fact reduce DOC delivery to Arctic streams, is timely and should be of interest to permafrost biogeochemistry researchers in general. I have some minor concerns detailed below, but I don't think they will greatly impede publication of this manuscript. The manuscript is well written, and the tables and figures are well presented and generally very clear.

[Figure]

One general comment is, and this could be addressed in the discussion for example without necessarily the need for extra data, what is the significance of this main finding to the overall carbon cycle/budget for such landscape features? The authors measure total suspended sediment (TSS) loss from the study features, but give no indication of the carbon content (I guess it wasn't measured). If DOC export is reduced due to adsorption to fine-grained sediments, these sediments are also mobilised and exported, and must carry some carbon. In Figure 3A we see that TSS increases downstream of these thaw features (unlike DOC), so can anything be said about the fate of the carbon that is locked up in this flux?

» Specific comments:

L. 17 (and/or introduction L. 66-69) – I recommend defining retrogressive thaw slumps specifically early on, do they differ from a normal thaw slump (active layer detachment, slide for example), and is the "retrogressive" characteristic of this type of slump especially different from thermokarst processes in general?

L. 49-51 – some resilience in the region is also possible in response to gradual permafrost thaw (e.g. Dean et al. 2016 doi: 10.1007/s10533-016-0252-2).

L. 53-56 – is DOC the primary substrate in soils or during aquatic transport/storage? I don't think there is a clear consensus on this point, and it's not clear exactly what your point is in this sentence. Yes, DOC can degrade to produce $CO_2$ (and $CH_4$) in Arctic aquatic environments, and this has been highlighted by recent studies (e.g. Spencer et al. 2015 GRL; Drake et al. 2015 PNAS; Mann et al. 2015 Nat Comms). But that doesn't mean DOC in the aquatic zone is the primary source of $CO_2$ released from streams and lakes. Most of the $CO_2$ released from streams is generated in the soil zone and transported laterally (Hotchkiss et al. 2015 Nat Geosci; Marx et al. 2017 doi: 10.1002/2016RG000547. So, I think it's important to not be too throw-away with this point, and be a bit clearer about how this aspect of Arctic aquatic carbon cycling relates to the study presented here.

[Figure]

L. 86-89 – would be good to compare this to pan-Arctic thermokarst estimates (e.g. Olefeldt et al. 2016 doi: 10.1038/ncomms13043).

L. 110-112 – please provide a reference to support this, or make it clear that this statement is hypothesis at this stage.

L. 126 – please be quantitative and give an area estimate, rather than saying "large portions" of the Arctic. Earlier the authors emphasise that the Peel Plateau is different, hence the uniqueness of this study. Please clarify the aims and intent regarding this point.

L. 180 – so these were exceptionally wet years? ~100 mm greater than the monthly averages? What is the significance of this enhanced precipitation to the DOC and TSS dynamics described in this study compared to other years?

L. 193 – how were the sites chosen to be representative? Was this done with remote sensing, or based on the experience of the authors? Either is fine, just to clarify.

L. 207 – suggest change to "... geomorphology were previously described by..." Or is the Malone et al. data actually used here?

L. 334-335 – would be good to clarify here and in the discussion, that only DOC concentrations are considered here, not fluxes. Dilution from the high rainfall reported for the study period could play a part. That said, the flux values from the FM3 site in Fig. 6 support the findings as presented.

Section 4.2 – might be useful for compare the geochemistry to pristine waters from the NWT (e.g. Dean et al. 2016 doi: 10.1007/s10533-016-0252-2).

L. 346 – I wouldn't consider $Ca^{2+}$ or $Mg^{2+}$ to be conservative in general, it would be good to justify whether they are conservative in the current study system.

L. 387-392 – with a rough mass balance, would it be possible to estimate the downstream $DO^{14}C$ values?

L. 399-401 – this is a very important point, showing that the fluxes show the same pattern as concentrations (see comments on L. 334-335). Maybe emphasize this linkage of the FM3 findings to all the study sites.

L. 421 – the Drake et al. and Mann et al. 2015 citations don't really fit here. These studies focus on degradation dynamics not mobilisation. Also, the Drake study used permafrost soils from Alaska. Please check the other references in this sentence.

L. 433-435 – DOC:ion ratios are key here to support the DOC flux vs. concentration issue I pointed out earlier. Why not emphasize these ratios more? (note comment regarding L. 346).

L. 439 – what field evidence specifically?

L. 447 – why not data from the other sites too?

L. 449-454 – much of this should really go in the methods.

L. 452 – TSS shouldn't be considered conservative in my opinion: entrainment and deposition processes would be occurring, so how can conservative behaviour be justified? Or is that the point here, that it isn't conservative? Hard to follow the reasoning here. Perhaps if the approach used here were more clearly outlined in the methods section, and the results presented more clearly (e.g. in a table), this might help clarify this point.

L. 469 – where is it sequestered? Within the exported TSS load, or in the depositional environments within the study systems? If sorbed to mineral complexes, is the carbon sill bioavailable? What does this process mean for the overall fate of the carbon released from the thaw slump features? This is an important aspect of the discussion that is currently missing.

L. 491 – what do these values actually say about the condition of the DOM? This could also be added to the results section.

L. 504-511 – would it be possible to quantify the relative contributions from these end members (e.g. Winterfeld et al. 2015 doi: 10.5194/bg-12-3769-2015)? Would make a nice addition to the manuscript.

L. 565 – "across multiple measurement points" – not all points were used?

L. 579-585 – this sentence begins with "This result clearly highlights…" but the rest of the sentence is long and unclear. Please rewrite to clarify the sentence's clarity.

L. 582-583 – Where is this evidenced in the present study?

L. 597 – inter-regional or slump type? If referring to the results of the present study vs. previous/pan-Arctic findings, then state this more definitively.

L. 602-606 – how important is thaw slump derived DOC to the Arctic carbon budget? Is it large enough to justify its inclusion in ESMs at this stage? Or is its inclusion into smaller-scale process models more justified? This information could also be incorporated into the introduction for context.

L. 610 – is this explored in the present study?

» Technical corrections:

L. 42-43 – it's good practice to cite specific chapters in the IPCC reports rather than the whole report or the summaries as used here.

L. 49 – remove the "s" from "thaws"

L. 93 – Should this be "thaw-" or "thermokarst-" affected regions, rather than "permafrost-affected regions"?

L. 141 – how brief is brief? Please give numbers.

L. 225 – Helicopter! Cool.

L. 232-240 – how long was there between sampling and analysis, generally?

L. 257-265 – please provide precision/sensitivity estimates for each analysis.

L. 300 – define AIC acronym when used for the first time.

L. 423 – missing space between comma and permafrost.

L. 457-460 – reference?

L. 483 – I would add the following references from the region near the current study: Street et al. 2016 doi: 10.1002/2016JG003387; Quinton and Pomeroy 2006 doi: 10.1002/hyp.6083).

L. 586 – remove semi-colon.

L. 612 – missing an "as"

L. 612 – should be Vonk et al. 2015b?

L. 619 – suggest changing "effects" to "impacts" otherwise too many effects/affects in this sentence.

Table 2. Is the "t" from the Wilcoxon output?

Please add the figure captions to the figures next time (personal preference).

Figure S1. These pictures are impressive, why not include in the main manuscript? Is it possible to include a rough scale in the photos also?

———————————————

---

## Referee Comment (RC2) · M. Fritz (Referee) · 31 Jul 2017

The manuscript provided by Bulger et al. investigates the role of retrogressive thaw slumps in moderating dissolved organic carbon delivery to stream ecosystems in north-western Canada. Comparing the similarities and differences in biogeochemical parameters upstream of such slumps, within slumps and downstream of slumps the authors convincingly conclude that adsorption processes between fine-grained mineral surfaces and DOC occur. The study shows that DOC gets removed from solution on short timescales and along short pathways, which is an important finding if we think about potential organic carbon mineralization into greenhouse gases and the potential destination of permafrost organic matter after thaw and mechanical mobilization. A minor shortcoming of study is that it touches inorganic hydrochemistry (e.g. major cations) but it does not use these data in the statistics for hypothesis testing - or I have missed this. In a next step I can see the potential in analyzing organo-mineral complexes and mineral surfaces upstream and downstream for validation of the different results in Canada, Alaska and Siberia.

The authors present original data and provide a very thorough and detailed description of the methods. In general, this topic and the presented data are of interest for researchers studying thermokarst, rapid permafrost degradation and arctic biogeochmistry. The language is generally very good and the figures and tables usefully complement the text.

I suggest the manuscript to be accepted after minor revisions.

**General comments:**

*Title:*
I suggest to change the title into:
"Retrogressive thaw slumps moderate dissolved organic carbon delivery to streams of the Peel Plateau, NWT, Canada"
This would highlight the process-driven character of biochemical interaction between DOC and . . .

The Introduction has 4 manuscript pages is therefore very long. Please cut (I have made some suggestions in the annotated manuscript attached) and align the internal structure according to the following points:
1. global relevance of the topic
2. specific relevance to the research field
3. previous work in this direction

4. knowledge gap(s)

5. overall aim how to fill the knowledge gap

6. objectives (specific and measurable)

**Specific comments:**

For specific comments see the annotated and attached pdf-file.

*Michael Fritz* (Alfred Wegener Institute, Helmholtz Centre for Polar and Marine Research)

Please also note the supplement to this comment:
https://www.biogeosciences-discuss.net/bg-2017-217/bg-2017-217-RC2-supplement.pdf

**Supplement:**

[revised manuscript text omitted]

---

## Author Comment (AC1) · 22 Sep 2017

This manuscript represents an important step in our understanding of permafrost thaw dynamics from the Canadian Arctic. As the authors point out, the permafrost regions of the Arctic are not all similar, and region-specific work such as this are critical to our understanding of the impact of climate change on the Arctic as a whole. The key finding of the manuscript, that permafrost thaw slumps may in fact reduce DOC delivery to Arctic streams, is timely and should be of interest to permafrost biogeochemistry researchers in general. I have some minor concerns detailed below, but I don't think they will greatly impede publication of this manuscript. The manuscript is well written, and the tables and figures are well presented and generally very clear.
*We thank Reviewer 1 for these positive comments, and for the detailed review. We have incorporated almost all of the reviewer suggestions, as detailed below.*

*Note that the 'current line numbers' listed below refer to line numbers in the uploaded, amended manuscript document. Because of the tracked changes in this document, there are fairly significant gaps in line numbers between some pages.*

One general comment is, and this could be addressed in the discussion for example without necessarily the need for extra data, what is the significance of this main finding to the overall carbon cycle/budget for such landscape features? The authors measure total suspended sediment (TSS) loss from the study features, but give no indication of the carbon content (I guess it wasn't measured). If DOC export is reduced due to adsorption to fine-grained sediments, these sediments are also mobilised and exported, and must carry some carbon. In Figure 3A we see that TSS increases downstream of these thaw features (unlike DOC), so can anything be said about the fate of the carbon that is locked up in this flux?
*Thanks for this good suggestion. We do see significant particulate organic carbon mobilization from these features; documenting this mobilization and the fate of this particle-attached carbon is currently underway, and being studied by another graduate student in our research group. While we have not added POC data to the current manuscript, we have added several lines of text to acknowledge the importance of particle-associated carbon flux (current L635-683).*

**Specific comments:**

L. 17 (and/or introduction L. 66-69) – I recommend defining retrogressive thaw slumps specifically early on, do they differ from a normal thaw slump (active layer detachment, slide for example), and is the "retrogressive" characteristic of this type of slump especially different from thermokarst processes in general?
*Agreed. We have added a clearer definition of retrogressive thaw slumps in the Introduction (current L86-87, including a reference to the image in Fig. 1), but not in the abstract, given space constraints.*

L. 49-51 – some resilience in the region is also possible in response to gradual permafrost thaw (e.g. Dean et al. 2016 doi: 10.1007/s10533-016-0252-2).
*Agreed. We have softened our wording in this sentence (current L54-56).*

L. 53-56 – is DOC the primary substrate in soils or during aquatic transport/storage? I don't think there is a clear consensus on this point, and it's not clear exactly what your point is in this sentence. Yes, DOC can degrade to produce $CO_2$ (and $CH_4$) in Arctic aquatic environments, and this has been highlighted by recent studies (e.g. Spencer et al. 2015 GRL; Drake et al. 2015 PNAS; Mann et al. 2015 Nat Comms). But that doesn't mean DOC in the aquatic zone is the primary source of $CO_2$ released from streams and

lakes. Most of the CO2 released from streams is generated in the soil zone and transported laterally (Hotchkiss et al. 2015 Nat Geosci; Marx et al. 2017 doi: 10.1002/2016RG000547. So, I think it's important to not be too throw-away with this point, and be a bit clearer about how this aspect of Arctic aquatic carbon cycling relates to the study presented here.

*Fair comment! The point we are trying to make here is a simple one – namely that DOC is a useful focal point when thinking about permafrost thaw and carbon, because it is the primary substrate for microbial mineralization. We've modified the sentence to be more inclusive of soil pore waters and the full aquatic continuum, by changing the text and removing the reference to Spencer (current L59-63).*

L. 86-89 – would be good to compare this to pan-Arctic thermokarst estimates (e.g. Olefeldt et al. 2016 doi: 10.1038/ncomms13043).

*At this point in the text, we are referring to individual thaw slumps ("individual thaw slumps commonly impact tens of hectares of terrain …"). We have modified our text slightly to clarify this point (L105-106), and have also made modifications towards the end of this paragraph (current L122-123) to state that "this till-associated, RTS-susceptible landscape type is found across the Laurentide and Barents-Kara glacial margins of Canada, Alaska, and Siberia".*

*We do not reference the Olefeldt paper here, because it provides a broad estimate for "hillslope thermokarst", which includes not only retrogressive thaw slump features, but also active layer detachments and thermal erosion gullies. However, we do now include this reference towards the end of our Discussion section.*

L. 110-112 – please provide a reference to support this, or make it clear that this statement is hypothesis at this stage.

*We have moved the Kothawala reference up from the sentence below to provide a clear citation for this statement.*

L. 126 – please be quantitative and give an area estimate, rather than saying "large portions" of the Arctic. Earlier the authors emphasise that the Peel Plateau is different, hence the uniqueness of this study. Please clarify the aims and intent regarding this point.

*We do not provide an area estimate (there isn't one available! See response above to L86-89 comment). However, we have modified the text to be much more specific, and read: "glacial margin landscapes throughout Canada, Alaska, and Siberia" (current L176). Based on this comment, we have also made slight modifications at other points in the introduction, to convey our meaning that the Peel Plateau is different from places where permafrost carbon release has been studied previously, rather than unique from a pan-Arctic perspective (see, for example, current L161-164). Thanks for helping us to clarify this important point.*

L. 180 – so these were exceptionally wet years? ~ 100 mm greater than the monthly averages? What is the significance of this enhanced precipitation to the DOC and TSS dynamics described in this study compared to other years?

*2014 was a wet year, although the recent trend on the Peel Plateau has been for exceptionally wet weather over the past ~decade, when compared to long-term norms. However, in double-checking this data, we noted that the comparison between 2014 and long-term average data was made between a (recently established) station on the Peel Plateau (for 2014), and a longer-term station at Fort McPherson (for the long-term average). We have now clarified this in the text, and added 2014 data for the Fort McPherson station. 2014 is still wetter than average, but the difference is not as substantial as it originally appeared (current L265-271).*

L. 193 – how were the sites chosen to be representative? Was this done with remote sensing, or based on the experience of the authors? Either is fine, just to clarify.
*Sites were chosen via an initial aerial (helicopter) survey, and previous knowledge of this landscape. We have modified the text to clarify this point (current line 275-276).*

L. 207 – suggest change to "...geomorphology were previously described by..." Or is the Malone et al. data actually used here?
*Modified as suggested. We do not use the Malone data in this paper; this sentence is merely intended to acknowledge this prior work.*

L. 334-335 – would be good to clarify here and in the discussion, that only DOC concentrations are considered here, not fluxes. Dilution from the high rainfall reported for the study period could play a part. That said, the flux values from the FM3 site in Fig. 6 support the findings as presented.
*We certainly recognize the importance of differentiating between DOC concentration and flux, and have made significant efforts to ensure that we are clear about when we are discussing one vs. the other. Towards this end, we have specified "DOC concentration" in the header for this section (4.1), and specify in this sentence that "DOC concentrations tended to be lower …".  To address this point in the discussion, we have added some text to our first Discussion section that specifically reiterates the flux (vs. concentration) results (current L587-588), and worked to clearly present (and demarcate) flux and concentration results in section 5.4.*

Section 4.2 – might be useful for compare the geochemistry to pristine waters from the NWT (e.g. Dean et al. 2016 doi: 10.1007/s10533-016-0252-2).
*Thanks for pointing us to this useful reference. We have chosen not to include it here, because the Trail Valley Creek region is somewhat different from the Peel Plateau (more organic-rich soils; greater prevalence of ice-wedge polygons; north of treeline), as evidenced by the significantly greater DOC concentrations across streams sampled around Trail Valley Creek, when compared to on the Peel Plateau. Because our 'upstream' sites present values for pristine waters in the Peel Plateau region, we thought it was best to keep the focus of this section on differences in DOC concentration between thaw-affected and pristine sites on the Peel Plateau, specifically.*

L. 346 – I wouldn't consider Ca2+ or Mg2+ to be conservative in general, it would be good to justify whether they are conservative in the current study system.
*Fair comment!  We have changed "conservative ions" to "major ions" here and elsewhere. Our intent for measuring Ca and Mg in this system was to track the impact of slumping, because we knew (at the outset of the study) that RTS features in this region had been shown to have significant concentrations of ions in their outflow waters.*

L. 387-392 – with a rough mass balance, would it be possible to estimate the downstream DO14C values?
*We don't undertake this type of analysis because samples for DO$^{14}$C were collected two years after the main study, to allow us to include this type of information in the paper. While it is unfortunate that our DO$^{14}$C samples are not true 'paired' samples with the DOC concentration data that is the focus of the paper, we feel they still contribute important information that aids with our understanding of the system.  As a result of the offset in collection date, we keep our analysis of the $^{14}$C data fairly simple.*

L. 399-401 – this is a very important point, showing that the fluxes show the same pattern as concentrations (see comments on L. 334-335). Maybe emphasize this linkage of the FM3 findings to all the study sites.

*Thanks for this comment – we agree that this is an important finding. We haven't added text to explicitly link the FM3 finding to all there study sites here (in the Results section), but do add some text towards this end in the Discussion (current L768-769), and in the methods, where we now outline the fact that FM3 was chosen for the more intensive flux work because it is roughly "representative of active slumps on the Peel Plateau with headwalls that erode Holocene- to Pleistocene-aged sediments" (current L362-363).*

L. 421 – the Drake et al. and Mann et al. 2015 citations don't really fit here. These studies focus on degradation dynamics not mobilisation. Also, the Drake study used permafrost soils from Alaska. Please check the other references in this sentence.

*Agreed that the Drake and Mann references are misplaced here. We have removed Drake, replaced the Mann reference with a reference to Spencer et al. (2015), re-worded this sentence slightly for clarity, and also removed a couple other references that were not entirely appropriate here (current L581-582).*

L. 433-435 – DOC:ion ratios are key here to support the DOC flux vs. concentration issue I pointed out earlier. Why not emphasize these ratios more? (note comment regarding L. 346).

*Thanks for this suggestion. We have added text to the results (current L504-508) and at this point in the Discussion (current L596-597) to further emphasize DOC: ion (and also DOC: TSS) ratios. We add these as inverse ratios (ie, ion:DOC) to more clearly demonstrate differences between upstream and downstream sites.*

L. 439 – what field evidence specifically?

*This statement refers to the evidence presented in the following few sentences. Text has been changed to "our results", for clarity (current L602).*

L. 447 – why not data from the other sites too?

*Our 'environmental controls' work at site FM3 provided us with the discharge data necessary for these calculations. The calculations specified in this section of the text were not planned – they were undertaken as a* post hoc *assessment to help us understand the patterns we were seeing a bit better. As a result, we're only able to do these calculations at site FM3, and not elsewhere. This is now clarified in the Methods section (current L458-459).*

L. 449-454 – much of this should really go in the methods.

*We have added details on these calculations to the methods (bottom of Section 3.4), and have deleted the text specifically dealing with the methodological component of this text from this section (current L 623-626).*

L. 452 – TSS shouldn't be considered conservative in my opinion: entrainment and deposition processes would be occurring, so how can conservative behaviour be justified? Or is that the point here, that it isn't conservative? Hard to follow the reasoning here. Perhaps if the approach used here were more clearly outlined in the methods section, and the results presented more clearly (e.g. in a table), this might help clarify this point.

*Thanks for this comment. We have changed our text here (current L624), and added text to the methods (bottom of Sn 3.4) to clarify our approach, which was to calculate a mass balance for DOC upstream and downstream of slump FM3, and couple this with a similar mass balance for TSS. Our calculations show*

*that the TSS load balances nicely (ie, downstream = upstream + inputs from slumping), while the DOC load decreases downstream of the slump, despite considerable DOC input from the slump feature. We've changed our wording to acknowledge that TSS is a "rough" (rather than conservative) tracer of slump inputs, given that entrainment and deposition are certainly occurring. We also add text to clarify that these calculations occur "over a < 1km span between upstream and downstream locations" (current L 462-463).*

L. 469 – where is it sequestered? Within the exported TSS load, or in the depositional environments within the study systems? If sorbed to mineral complexes, is the carbon sill bioavailable? What does this process mean for the overall fate of the carbon released from the thaw slump features? This is an important aspect of the discussion that is currently missing.
*Our hypothesis at this time is that the dissolved OC becomes sequestered on mineral surfaces. Although we do not know whether this carbon is still bioavailable (there is very little in the literature discussing decomposition of POC!) this is something other members of our research group are actively working on. We've clarified our text to explain that we expect that this sequestration is occurring onto mineral surfaces, and have also added text to discuss POC fluxes in general from these slump features (see also our response to the general comments by this referee, above; current L635 - 682).*

L. 491 – what do these values actually say about the condition of the DOM? This could also be added to the results section.
*We have added some text to describe what these values indicate for DOM character (current L699-700). This text is already present in the results section (Section 4.3, current L511 and 516), so we have not modified the text there.*

L. 504-511 – would it be possible to quantify the relative contributions from these end members (e.g. Winterfeld et al. 2015 doi: 10.5194/bg-12-3769-2015)? Would make a nice addition to the manuscript.
*We agree that this would have been a wonderful addition to the manuscript. However, we're not really able to undertake this kind of analysis because the samples are not paired. See also our response to the comment for lines 387-392.*

L. 565 – "across multiple measurement points" – not all points were used?
*We did use all of the data here: "across the multiple measurement points that we considered". To increase clarity, we've removed this component of the sentence (current L802).*

L. 579-585 – this sentence begins with "This result clearly highlights … " but the rest of the sentence is long and unclear. Please rewrite to clarify the sentence's clarity.
*Fair comment! This has been split into two sentences and reworded for clarity (current L817-822).*

L. 582-583 – Where is this evidenced in the present study?
*We think the reviewer is referring to the statement that slump features "drain contemporary active layers". Because thaw slump headwalls expose the contemporary active layer, in addition to Holocene and Pleistocene layers (see Figure 1c and d), drainage from these features necessarily incorporates active layer materials. Hopefully our re-working of this long sentence (see comment and response for L579-585 above) also helps to clarify our text.*

L. 597 – inter-regional or slump type? If referring to the results of the present study vs. previous/pan-Arctic findings, then state this more definitively.

*We have re-worked the text in this section somewhat for clarity (current L851-856 and onwards). In this sentence, and the sentences that follow, we are making the point that we should expect permafrost thaw (and, it's biogeochemical consequences) to play out differently in different regions of the Arctic. We have changed our wording slightly to link this sentence more clearly to the one that follows it, in which we specifically compare our results to findings from other regions.*

L. 602-606 – how important is thaw slump derived DOC to the Arctic carbon budget? Is it large enough to justify its inclusion in ESMs at this stage? Or is its inclusion into smaller-scale process models more justified? This information could also be incorporated into the introduction for context.
*Our sentence here was considering more the fact that DOC release as a result of thermokarst (or, permafrost thaw generally) could be predicted somewhat with a consideration of factors such as Quaternary history, soil composition, and the nature of permafrost thaw. This, in-turn, could lend itself to modelling efforts to understand variability in the effects of thermokarst at the pan-Arctic scale. To address the reviewer's comment, we have modified our text in this sentence slightly, including changing the beginning of the sentence from "modelling efforts" to simply "efforts" (current L856-859).*

L. 610 – is this explored in the present study?
*I think this question comes from a change in terminology: we use "paleo active layer" here to refer to the Holocene-aged layer we discuss throughout the manuscript. Our text has been modified to "relative depth of the Holocene-aged paleo active layer" (current L864).*

**Technical corrections:**

L. 42-43 – it's good practice to cite specific chapters in the IPCC reports rather than the whole report or the summaries as used here. *Done.*

L. 49 – remove the "s" from "thaws". *Done.*

L. 93 – Should this be "thaw-" or "thermokarst-" affected regions, rather than "permafrost-affected regions"? *"Permafrost" has been changed to "thaw".*

L. 141 – how brief is brief? Please give numbers. *We now specify "a maximum of 2,000-3,000 years" in the text.*

L. 225 – Helicopter! Cool. *Thanks! We definitely enjoyed this part of the study …*

L. 232-240 – how long was there between sampling and analysis, generally? *This detail has been added (current L355-357).*

L. 257-265 – please provide precision/sensitivity estimates for each analysis. *Precision or error estimates have been added as applicable throughout this paragraph (current L383 and onwards).*

L. 300 – define AIC acronym when used for the first time. *Done.*

L. 423 – missing space between comma and permafrost. *Space has been inserted.*

L. 457-460 – reference? *The reference had been provided at the end of the following sentence; it has been moved up for clarity.*

L. 483 – I would add the following references from the region near the current study: Street et al. 2016 doi: 10.1002/2016JG003387; Quinton and Pomeroy 2006 doi: 10.1002/hyp.6083). *We have added the Street reference here, but have not added Quinton and Pomeroy because it focuses on major ion chemistry, rather than organic constituents.*

L. 586 – remove semi-colon. *Done.*

L. 612 – missing an "as". *Inserted.*

L. 612 – should be Vonk et al. 2015b? *The Watanabe reference at the end of this sentence is as intended. The Watanabe study looks at within-region variations in lake chemistry in response to thermokarst. We like this specific study for the argument we are presenting in this sentence, rather than the more general review presented in Vonk et al. 2015.*

L. 619 – suggest changing "effects" to "impacts" otherwise too many effects/affects in this sentence. *We avoid this use of 'impacts' because it has a specific meaning in the geophysical sciences, but have re-worded this sentence to avoid the (substantial!) repetition (current L896 onwards).*

Table 2. Is the "t" from the Wilcoxon output? *The t is from the mixed effects model. This is now clarified in the Figure caption.*

Please add the figure captions to the figures next time (personal preference). *Figure captions are included with figures for this re-submission.*

Figure S1. These pictures are impressive, why not include in the main manuscript? Is it possible to include a rough scale in the photos also? *We put a fair bit of consideration into this request, but have decided to leave the images in the Appendix because (1) they are somewhat repetitive with panels c-e in Figure 1 and (2) to avoid an extra page of manuscript length. We haven't added a scale bar because we didn't take measurements that would allow us to do this when we took the photographs. However, the figure caption does point the reader to Table 1, which provides aerial estimates for each of the slumps that we studied.*

---

## Author Comment (AC2) · 22 Sep 2017

**Referee 2: Micha Fritz**

The manuscript provided by Bulger et al. investigates the role of retrogressive thaw slumps in moderating dissolved organic carbon delivery to stream ecosystems in north-western Canada. Comparing the similarities and differences in biogeochemical parameters upstream of such slumps, within slumps and downstream of slumps the authors convincingly conclude that adsorption processes between fine-grained mineral surfaces and DOC occur. The study shows that DOC gets removed from solution on short timescales and along short pathways, which is an important finding if we think about potential organic carbon mineralization into greenhouse gases and the potential destination of permafrost organic matter after thaw and mechanical mobilization.

A minor shortcoming of study is that it touches inorganic hydrochemistry (e.g. major cations) but it does not use these data in the statistics for hypothesis testing - or I have missed this. In a next step I can see the potential in analyzing organo-mineral complexes and mineral surfaces upstream and downstream for validation of the different results in Canada, Alaska and Siberia.

The authors present original data and provide a very thorough and detailed description of the methods. In general, this topic and the presented data are of interest for researchers studying thermokarst, rapid permafrost degradation and arctic biogeochmistry. The language is generally very good and the figures and tables usefully complement the text.

I suggest the manuscript to be accepted after minor revisions.
*We thank reviewer 2 for these positive comments, and also for the very thorough review of our manuscript. A brief response to the comment on our major cation data: we present (and collected) these data to bolster this work, which was designed to focus on dissolved organic carbon release from slumps. As a result, these data are presented in an ancillary fashion. However, we do incorporate them into our statistical analyses (see Table 2). Although the cations are not a focus of the current manuscript, we certainly agree that this study has uncovered organo-mineral interactions as a fruitful direction for future study: both in this region, specifically, and for cross-region comparisons.*

**General comments:**

Title: I suggest to change the title into: "Retrogressive thaw slumps moderate dissolved organic carbon delivery to streams of the Peel Plateau, NWT, Canada". This would highlight the process-driven character of biochemical interaction between DOC and ...
*We use 'temper' as : "Act as a neutralizing or counterbalancing force to (something)" [Oxford dictionary], and prefer this to 'moderate' in the title. We've kept the title as-is, rather than switching 'temper' to 'moderate'. Thanks for this suggestion, however!*

The Introduction has 4 manuscript pages is therefore very long. Please cut (I have made some suggestions in the annotated manuscript attached) and align the internal structure according to the following points:
1. global relevance of the topic
2. specific relevance to the research field
3. previous work in this direction
4. knowledge gap(s)
5. overall aim how to fill the knowledge gap
6. objectives (specific and measurable)

*Thanks for this useful comment. We have worked to streamline the Introduction by: making the majority of changes suggested in the annotated manuscript (see further details below); adding a 'knowledge gap' statement (current L165-169); clarifying the last paragraph to include a clear statement of objectives (current L170 onwards; see uploaded PDF); and making some additional editorial changes to streamline the text. We have retained two paragraphs in the Introduction that discuss retrogressive thaw slumps: both generally (paragraph 2), and on the Peel Plateau specifically (paragraph 3). While we have streamlined these paragraphs somewhat, they do add a fair bit of bulk beyond a 'normal' Introduction. However, we feel that this text is important to include in the Introduction, because this feature type may not be familiar to biogeochemists that study permafrost thaw, and understanding the Peel Plateau landscape is critical for understanding the study objectives.*

**Specific comments:**

For specific comments see the annotated and attached pdf-file.
*Thanks for this file. Below we've listed line numbers where modifications were not made as requested in the annotated PDF, or where we felt some additional comment might be useful. Except for the items listed below, all changes were made as suggested. Line numbers in bold below refer to the commented PDF submitted by Dr. Fritz.*

-**line 24:** *Regarding question about whether the process we discuss is caused by adsorption to clays.*
This would certainly be our hypothesis! However, at this point, this has not been explicitly tested, so we have not added additional text to the abstract.

-**line 24-27** is highlighted but I can't find a comment associated with this text, so no changes were made.

-**line 44:** "forecast" (rather than "forecasted") was retained; as a result we also haven't made the other grammatical changes in this sentence.

-**line 70:** We add Lantuit et al., (2012) but not Lantuit and Pollard (2006); the Kokelj reference provides a broad perspective on thaw slumps from across the western Canadian Arctic, so we've chosen to add only one reference from Herschel Island at this point in the text, to also add this more specific perspective.

-**line 77:** We keep "materials", rather than "material", and therefore have also not modified the remainder of this sentence (to the singular).

-**line 82:** an insertion is indicated, but I could not see text associated with this, so no change was made.

-**line 355:** *Regarding the following comment: This also means that thawing ice-rich Pleistocene strata which contain large ground ice contents dilute the DOC signal from organic-rich horizons (see Fritz et al., 2015 [The Cryosphere] and Tanski et al. 2016 [GBC] for typical DOC concentrations in different ground ice types).*
Agreed! We haven't modified the text here (to avoid adding interpretation or discussion of results to the Results section), but have added text to clarify the importance of ground ice later on in the Discussion, including a specific reference to these two citations (current L743-744).

-**line 342 (Section 4.2):** While we added some DOC values as suggested for sections 4.1 and 4.4, we haven't added major ion data here. We've chosen to keep the focus of the text in this section on relative change in major ion concentration between upstream and downstream sites.

-**line 438:** *Regarding the comment that the 'dilution effect' or low DOC concentrations that we see for slump outflow from some slumps is caused by low DOC ground ice.*
This ties to our response to the comment about L355 above. As mentioned above, we agree that this is worth clarifying in our text. In this section (5.1) we are discussing more patterns in DOC across slumps; the mechanisms for these patterns are more thoroughly dealt with in section 5.3. To address this comment, we have pointed the reader to Section 5.3 at this point in the text (current L601), and added a specific consideration of DOC and ground ice in that section (current L743-744), including reference to the Fritz and Tanskii citations.

-**line 462:** *Regarding the comment "This sentence does not provide much useful information and should be somehow combined with the next sentence to carve out the differences of the landscapes and stratigraphy".*
This section of the Discussion has been re-worked slightly, in part in response to comments that were also made by Reviewer 1. We split this sentence in two to allow the second half to tie more clearly to the sentence that follows (current L626-629).

-**line 461:** "Lentic" was retained, rather than switching to "limnic"

-**line 552**: *Regarding additional hypothesis about the DOC: temperature link being caused by temperature effects on headwall thaw.*
This point in the text discusses the relationship between temperature and DOC at upstream (i.e., pristine) sites. As a result, we have not added this hypothesis, which is applicable only to slump-impacted sites. We do discuss our findings for slump-impacted sites in the following paragraph of the manuscript. To address the reviewer's point, we have modified our wording to clarify that this component of the text deals with upstream-of-slump sites only (minor changes for clarity throughout the first paragraph of Sn 5.4).

-**line 611-614**: text is highlighted, but no comment is present, so no change was made.

-**line 615**: we retain "for example" to allow us to indicate that it's not just the Peel Plateau that is different from other regions of the Arctic.

-**line 617:** *Regarding the comment: "These are not the best references to active layer deepening and its effects. Look at the CALM papers in GTN-P"*
The Kokelj and Vonk citations are provided to reference the statement that "non-linearity can also be expected to extend to different types of permafrost thaw …" (ie, the finding of non-linear response). As a result, we have retained them here, but we also add a CALM paper (Romanovsky et al., 2010) to make sure we also include a reference that documents the process of active layer deepening, specifically.

-**Table 3:** *Regarding question about why only a subset of our slump features have 14C measurements.*
We went back to our sites two years after the main study to collect 14-C measurements to help us better refine this story. We only have measurements from 4 of our 8 slumps because some of these features are difficult to access (helicopter only).

---

## Author Comment (AC3) · 22 Sep 2017

[revised manuscript text omitted]

| 166 | mechanisms governing the thaw-mediated transport of DOC from land to freshwater seem likely to               |                   |                                       |
| 167 | differ in till-dominated landscapes when compared to other regions studied to date, little is known          |                   |                                       |
| 168 | about the downstream consequences of permafrost thaw for carbon biogeochemistry in regions such as           |                   |                                       |
| 169 | the Peel Plateau.                                                                                            |                   |                                       |
| 170 | The objective of this study is to quantify how RTS features affect the concentration and                     |                   | Deleted                               |
|     |                                                                                                              |                   | Deleted:                              |
| 171 | composition of DOC across a series of slump-affected streams on the Peel Plateau, and to examine how         |                   | Deleted                               |
| 172 | observed variation in slump morphology affects DOC dynamics in slump-affected downstream                     |                   | Deleted                               |
| 173 | environments. We further investigate how short-term variation in precipitation, temperature, and solar       |                   | Deleted:                              |
| 174 | radiation affect DOC delivery from land to water, using measurements of DOC flux above and below a           |                   |                                       |
| 175 | single RTS feature, We target the thermokarst-sensitive Peel Plateau for this work , which is         |                   | Deleted:                              |
| 176 | characteristic of till-rich, glacial margin landscapes throughout Canada, Alaska, and Siberia (Kokelj et al. | $\bigcirc$        | Deleted                               |
| 177 | 2017b). By comparing our results to those from other regions, this allows us to consider how broad           | $\langle \rangle$ | Deleted:
| 178 | variation in permafrost soil composition, permafrost genesis, and Quaternary history may drive variation     | $\swarrow$        | Deleted:                              |
| 179 | in Jand-freshwater DOC dynamics across divergent regions of the Arctic affected by permafrost thaw.          | $\square$         | Deleted:                              |
|     | ···· /                                                                                                       | (N)               | Deleted:                              |
| 180 |                                                                                                              | () ()             | Deleted:                              |
| 181 |                                                                                                              |                   | Deleted:                              |
| 101 |                                                                                                              |                   | Deleted:                              |
| 182 | 2 Study Site                                                                                                 | /                 | Deleted:                              |
| 183 | 2.1 General study site description                                                                           |                   | understan
terrain, an
temper th |
| 184 | Our study was conducted on the Peel Plateau, situated in the eastern foothills of the Richardson             |                   | interaction
landscape              |

| -{ | Deleted: Sorption      |
|----|------------------------|
| ,  |                        |
| 1  | Deleted: processes     |
| -{ | Deleted: transport     |
| 1  | Deleted: the glaciated |
| 1  | Deleted: landscape     |
| -( | Deleted: thermokarst   |
| 1  | Deleted: effectively   |

| 1 | Deleted: In this study, we        |
|---|-----------------------------------|
| ſ | Deleted: on the Peel Plateau      |
| - | Deleted: within                   |
| 1 | Deleted: recipient stream systems |

| Ι                         | Deleted: , to explore the drivers of temporal variation in DOC flux                                                                                                                                                                                                                                                                                                                                                                                                                                                                                          |
|---------------------------|---------------------------------------------------------------------------------------------------------------------------------------------------------------------------------------------------------------------------------------------------------------------------------------------------------------------------------------------------------------------------------------------------------------------------------------------------------------------------------------------------------------------------------------------------------------------|
|                           | Deleted: specifically                                                                                                                                                                                                                                                                                                                                                                                                                                                                                                                                               |
| $\square$                 | Deleted: se                                                                                                                                                                                                                                                                                                                                                                                                                                                                                                                                                         |
| $\langle \rangle$         | Deleted: glacial deposits                                                                                                                                                                                                                                                                                                                                                                                                                                                                                                                                           |
| $\langle \rangle$         | Deleted: are                                                                                                                                                                                                                                                                                                                                                                                                                                                                                                                                                        |
| ľ                         | Deleted: large portions                                                                                                                                                                                                                                                                                                                                                                                                                                                                                                                                             |
| Υ                         | Deleted: of the circumpolar Arctic, to explicitly                                                                                                                                                                                                                                                                                                                                                                                                                                                                                                            |
| $\square$                 | Deleted: s                                                                                                                                                                                                                                                                                                                                                                                                                                                                                                                                                          |
| Ŋ                         | Deleted: ,                                                                                                                                                                                                                                                                                                                                                                                                                                                                                                                                                          |
| //                        | Deleted: influence                                                                                                                                                                                                                                                                                                                                                                                                                                                                                                                                                  |
| Ν,                        | Deleted: variability                                                                                                                                                                                                                                                                                                                                                                                                                                                                                                                                                |
| $\langle \rangle \rangle$ | Deleted: permafrost-DOC interactions                                                                                                                                                                                                                                                                                                                                                                                                                                                                                                                                |
|                           | Deleted: vast                                                                                                                                                                                                                                                                                                                                                                                                                                                                                                                                                       |

[revised manuscript text omitted]
                        | Deleted: analysis                                                                                          |
| 224 | HDRE battles with no bookspace and cooled. During summer 2016, samples were additionally collected                          | Deleted: analyses                                                                                          |
| 334 | HDPE bottles with no headspace and sealed, During summer 2016, samples were additionally collected                          | Deleted: . Bottles were                                                                                    |
| 335 | from a subset of slump locations (FM2, FM3, FM4 and SD) for the 14 C signature of DOC at upstream and            | Deleted: and refrigerated until analysis                                                                   |
|     |                                                                                                                             | Deleted: Field                                                                                             |
| 336 | within-slump sites. DO14C samples were collected in acid -washed polycarbonate bottles, allowed to            | Formatted: Superscript                                                                                     |
| 337 | settle for 24 hours, and filtered using pre-combusted Whatman GF/F filters into pre-combusted glass                         | Deleted: pre                                                                                               |
| 557 | sectie for 24 hours, and intered using pre-combusted whatman of /1 inters into pre-combusted glass                          | Deleted: 1-2 L                                                                                             |

[revised manuscript text omitted]
 work to-date in other areas of the Arctic, where                                            | $\swarrow$   |
| 630 | thermokarst has been demonstrated to lead to an efflux of high-DOC waters from slump features (e.g.,                                                 |              |
| 631 | Abbott et al., 2014; Vonk et al., 2013a). Ice-marginal glaciated landscapes are common throughout the                                                |              |
| 632 | western Canadian Arctic, however, and in many other Arctic regions. This terrain type is characterized by                                            |              |
| 633 | thick, mineral-rich but carbon_poor tills, and high ice contents that are predisposed to intense climate-                                            |              |
| 634 | driven thaw slumping and the release of glacigenic sediments (Kokelj et al., 2017b). As a result, DOC                                                |              |
| 635 | 'sequestration' following slumping seems unlikely to be limited to the Peel Plateau. Given the high TSS                                              |              |
| 636 | export and apparent organic carbon sorption to glacigenic sediments observed with slumping on the                                                    |              |
| 637 | Peel Plateau, we expect that substantial organic carbon is mobilized from these slumps in the particle-                                              | 1            |
| 638 | attached, rather than dissolved, form (i.e., as particulate organic carbon; POC). Quantifying this POC                                               |              |
| 1   | 21                                                                                                                                                   |              |

|                | eleted: these                                                                                                                                                                                                                                                                                                                  |
|----------------|--------------------------------------------------------------------------------------------------------------------------------------------------------------------------------------------------------------------------------------------------------------------------------------------------------------------------------|
| D              | eleted: es                                                                                                                                                                                                                                                                                                                     |
| D              | Peleted: within                                                                                                                                                                                                                                                                                                                |
| D              | eleted: calculated as [DOC] within • (discharge down – discharge up );                                                                                                                                                                                                                 |
| D              | Peleted: conservative                                                                                                                                                                                                                                                                                                          |
| D              | eleted: activity                                                                                                                                                                                                                                                                                                               |
| D              | eleted: within-slump flux                                                                                                                                                                                                                                                                                                      |
| D              | eleted: (as [TSS] within • (discharge down – discharge up ))                                                                                                                                                                                                                                  |
| D              | eleted: difference                                                                                                                                                                                                                                                                                                             |
| D              | eleted: between                                                                                                                                                                                                                                                                                                                |
| D              | eleted: and upstream locations                                                                                                                                                                                                                                                                                                 |
| D              | eleted: is                                                                                                                                                                                                                                                                                                                     |
| D              | eleted: of slumps                                                                                                                                                                                                                                                                                                              |
| D              | Peleted: The                                                                                                                                                                                                                                                                                                                   |
| D              | eleted: downstream of Peel Plateau slumps                                                                                                                                                                                                                                                                                      |
| D              | eleted: is similar to, but more muted than results                                                                                                                                                                                                                                                                             |
| D              | eleted: ,                                                                                                                                                                                                                                                                                                                      |
| cl
co       | eleted: where following slump stabilization, lakes are haracterized by increases in conductivity, clear decreases in DOC oncentration, and a strong negative correlation between these wo parameters                                                                                                                    |
| is
ei
se | veleted: . The greater magnitude of effect for lakes in this region
likely caused by substantial particle settling in lentic
nvironments, which enables DOC scavenging with the inorganic
ediment inputs of thermokarst (Kokelj et al., 2005). Although
ecreasing DOC with RTS activity on the Peel Plateau |
| D              | eleted: s                                                                                                                                                                                                                                                                                                                      |
| D              | eleted: regions (e.g., Abbott et al., 2014; Vonk et al., 2013a),                                                                                                                                                                                                                                                               |
| D              | eleted: i                                                                                                                                                                                                                                                                                                                      |
| D              | eleted: intensely affected by RTS                                                                                                                                                                                                                                                                                              |
| D              | eleted: ,                                                                                                                                                                                                                                                                                                                      |
| D              | eleted: (Kokelj et al., 2017b). In general, t                                                                                                                                                                                                                                                                                  |
| D              | eleted: typically                                                                                                                                                                                                                                                                                                              |
| D              | eleted:                                                                                                                                                                                                                                                                                                                        |
| D              | eleted: which with their                                                                                                                                                                                                                                                                                                       |
|                | alatadi Thus, it seems likelu that                                                                                                                                                                                                                                                                                             |
| D              | eleted: Thus, it seems likely that                                                                                                                                                                                                                                                                                             |
| D              | eleted: Thus, it seems likely that
eleted: the processes we observe are not limited to the Peel
lateau: research to quantify                                                                                                                                                                                             |

[revised manuscript text omitted]

sorption processes seems warranted across regions karst intensifies the transport of mineral-rich ownslope aquatic systems.

/A254 and

pared to

[revised manuscript text omitted]

| 841 | resultant biogeochemical effects, is clearly warranted on the Peel Plateau and elsewhere, we must also                                                                                                                |                         | Deleted: ; both                                                                                                     |
|-----|-----------------------------------------------------------------------------------------------------------------------------------------------------------------------------------------------------------------------|-------------------------|---------------------------------------------------------------------------------------------------------------------|
|     |                                                                                                                                                                                                                       |                         | Deleted: across the Arctic                                                                                          |
| 842 | recognize that environmental controls on slump activity and thus downstream biogeochemistry can be                                                                                                                    | Ì                       | Deleted: where                                                                                                      |
| 843 | expected to show marked regional variation (see for example, work from Eureka Sound; Grom & Pollard

---

## Author Response (AR1)

**Response to Associate Editor Comments for BG-2017-217**

Comments to the Author:

Dear Suzanne, Cara,

Thank you for your detailed replies to both referee reports. I have now gone through both sets of author replies and the revised version of your manuscript, and feel you have thoroughly and convincingly addressed the suggestions and comments made. Considering also the favourable recommendations by both reviewers, I am therefore pleased to accept your revised version for Biogeosciences.
I would like to make 2 additional suggestions to consider before uploading the final version of your files:

-In the abstract, I feel the conclusions concerning the different water sources (best explored in section 5.3 but relevant not also for DOC concentrations) would be worth including more explicitly, i.e. a key message would be that different water sources contribute to the outflow of the slumps, that these have different DOC concentrations and characteristics (and age), but that the overall decrease in DOC concentrations is not simply a mixing of different sources but that other DOC losses (the proposed mechanism being sorption to particles) also occur.

-I would be strongly in favour (as for all manuscripts) to provide the full data as an electronic supplement. I feel the dataset should be relatively straightforward to organize in a way that is clear to potential users - and that making the full data publicly available will increase their potential value (e.g. use in larger data syntheses).

With best regards
Steven Bouillon

Dear Steven,

Thank you for your kind comments – we are excited to have our paper accepted with minor revisions in Biogeosciences!  In response to your comments, we have reworked the abstract in an attempt to more clearly outline the findings in Section 5.3.  We also now include our data in Supplemental Tables S1 and S2.

In addition to the changes outlined above, all authors have re-read the manuscript one last time, which has resulted in some editorial changes.  The updated manuscript has been submitted as a tracked changes PDF (attached below), and as a clean PDF.  The edits made in this round were all stylistic, and none of them affect the changes that were made in response to the reviewers' comments.

Best regards,

Suzanne Tank (on behalf of Cara Bulger, and Steve Koklelj)

[revised manuscript text omitted]

---

## Author Response (AR2)

Non-public comments to the Author:

Dear Suzanne, Cara,

Thank you for your final revisions, and I appreciate that you've added the full dataset as a supplement. For the latter, just one final suggestion: I have the impression there's a rather liberal number of decimals used for certain parameters, and not always consistent. I'll leave this up to you to decide where it might be appropriate to trim the numbers down to the significant ones.

Best regards
Steven

Dear Steven,

Thanks very much for your rapid response to our re-submission. In response to your comment, we have amended our supplementary datasets as follows:

**TSS and Conductivity**: presented with 1 decimal place for values below 1,000, and zero decimal places for values great than 1,000.
**DOC, ions, SUVA, $S_{275-295}$, $S_{350-400}$, and 18-O**: presented with 1 decimal place
**$S_R$**: presented with 2 decimal places.

Best regards,

Suzanne Tank and Cara Bulger